# In Search of Unambiguous Evidence of the Fulde–Ferrell–Larkin–Ovchinnikov State in Quasi-Low Dimensional Superconductors

**Mihail D. Croitoru** [1,] **and Alexandre I. Buzdin** [1,2,]

[1]  Université Bordeaux, CNRS, LOMA, UMR 5798, F-33405 Talence, France;
    mihail.d.croitoru@gmail.com (M.D.C.); alexander.bouzdine@u-bordeaux.fr (A.I.B.)
[2]  Institut Universitaire de France, F-75231 Paris, France

**Abstract:** In layered conductors with a sufficiently weak interlayer coupling in-plane magnetic field cause only small diamagnetic currents and the orbital depairing is strongly suppressed. Therefore, the Zeeman effect predominantly governs the spin-singlet superconductivity making the formation of the spatially modulated Fulde–Ferrell–Larkin–Ovchinnikov (FFLO) phase possible in such materials. Despite decades of strenuous effort, this state still remains a profound mystery. In the last several years, however, there have been observed several hints indicating the experimental realization of the FFLO state in organic layered superconductors. The emergence of the FFLO phase has been demonstrated mainly based on thermodynamic quantities or microscopically with spin polarization distribution that exhibit anomalies within the superconducting state in the presence of the in-plane magnetic field. However, the direct observation of superconducting order parameter modulation is so far missing. Recently, there have been proposed theoretically several hallmark signatures for FFLO phase, which are a direct consequence of its main feature, the spatial modulation of the order parameter, and hence can provide incontrovertible evidence of FFLO. In this article, a review of these signatures and the underlying theoretical framework is given with the purpose to summarize the results obtained so far, omitting duplications, and to emphasize the ideas and physics behind them.

**Keywords:** FFLO; superconducting phase transitions; low-dimensional systems

## 1. Introduction

High magnetic fields destroy superconductivity and restore the normal conducting state [1]. The underlying physics are based on two effects. The first one is orbital in nature whereby vortices penetrate into a superconductor and the suppression of superconductivity by a magnetic field $H$ is caused by an increase of diamagnetic energy beyond the critical value $H_{c2}^{orb} = \Phi_0/2\pi\xi^2$. The second one is the Zeeman effect, which breaks apart the paired electrons in a spin-singlet state if the field is larger than the Pauli paramagnetic limit, $\mu_B H_P = \Delta_0/\sqrt{2}$, where $\Delta_0$ is the superconducting gap at $T = 0$ and $H = 0$ [2,3]. In most type-II superconductors, the orbital effect plays the dominant role in suppression of superconductivity. However, the upper critical field becomes high when the orbital motion is suppressed like in quasi-1D or quasi-2D layered superconductors for magnetic field applied parallel to the most conducting layers. Under this condition, the destruction of spin-singlet state of Cooper pairs may become the dominant mechanism of the superconducting state suppression, with the upper critical field determined mostly by the Pauli paramagnetic limit [4].

Due to the paramagnetism of conduction electrons, in a magnetic field, the conduction band is splitted and the normal quasiparticles have separate spin-up and spin-down Fermi surfaces (illustrated in Figure 1). They are displaced due to the Zeeman energy. The Fermi surface mismatch should lead to a pair breaking: in the conventional Bardeen–Cooper–Schrieffer (BCS) pairing regime,

the total pair momentum is zero, a pairing state, Cooper pair, can be formed between any opposite parts of the Fermi surface. When magnetic field is above the Pauli limit, it is difficult to form pairs with zero total momentum because the difference in displaced Fermi energies becomes larger than the pairing gap. To overcome this, the Zeeman splitting causes a nonzero momentum of the Cooper pairs, resulting in oscillations of the superconducting order parameter, at the expense of the pairs being formed only between some restricted parts of the Zeeman splitted Fermi surfaces [5]. These parts depend on the anisotropy of the Fermi surface and determine the stabilization of the Fulde–Ferrell–Larkin–Ovchinnikov (FFLO) state (in addition to U(1)-gauge symmetry, the spatial symmetry is spontaneously broken). The existence of inhomogeneous superconducting state was predicted back in the 1960s by Fulde and Ferrell and Larkin and Ovchinnikov [6,7], who pointed out that the stability of the superconducting phase in a magnetic field can be increased beyond the Pauli paramagnetic limit by pairing electrons with different momenta. The FFLO emergence temperature cannot exceed $T^* \simeq 0.56T_{c0}$. However, the influence of impurities may reduce $T^*$ [8]. Physically spatial modulation of the order parameter allows for arranging the polarized quasiparticles in the vicinity of the nodes of the order parameter, and hence compensating for the loss of the superconducting condensation energy by the gain in spin polarization [5]. An FFLO state consists of regions of positive and negative pairing amplitude separated by domain walls where the magnetization is accumulated—a form of phase separation with alternating superconducting and polarized normal regions.

Conditions for the stabilization of the FFLO phase are rather stringent [9], namely (i) the orbital pair breaking effect should be sufficiently weaker than the Pauli paramagnetic limit, the Maki parameter $\alpha_M \equiv \sqrt{2}H_{c2}/H_P \gtrsim 1.8$; (ii) the system should be in a clean limit, $l \gg \xi_0 = \hbar v_F/\pi\Delta_0$, where $l$ is the mean-free path of the quasiparticles and $\xi_0$ is the superconducting coherence length at $T = 0$ and $H = 0$ [8,10–14]. Moreover, the highly anisotropic Fermi surface [5,15–17] favors the FFLO phase formation. This is an inherent property of layered conductors that exhibit a highly anisotropic structure and hence have features of systems with reduced dimensionality [18].

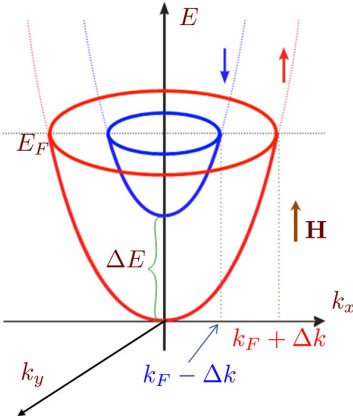

**Figure 1.** Schematic representation of the paramagnetism of conduction electrons. Zeeman effect.

However, despite intense theoretical and experimental efforts since its prediction in the 1960s, the compelling evidence for nonhomogeneous state in superconductors has been provided only recently. There have been observed several signatures indicating the experimental realization of the FFLO state in organic superconductors for in-plane external magnetic field when the flux penetrates between the layers in the form of Josephson vortices, thus limiting orbital suppression [19–21]. In particular, (i) the anomaly in the thermal conductivity [22], and fine details in the phase diagram obtained by the tunnel diode oscillator (rf)-penetration depth measurements and pulsed field techniques [23] for the clean organic sample $\lambda$-(BETS)$_2$GaCl$_4$; (ii) the magnetic torque evidence for the appearance of an additional first-order phase transition line within the superconducting phase in the in-plane high field regime for organic sample $\kappa$-(BEDT-TTF)$_2$Cu(NCS)$_2$ [24,25]; (iii) evidence for phase transition within the

superconducting phase obtained from the local electronic spin polarization in $^{13}$C NMR spectroscopy experiment [26], NMR detection of spin-polarized quasiparticles [27] forming the Andreev bound states spatially localized in the nodes of the order parameter [5,28,29], and phase transitions in the $(H - T)$ phase diagram that are consistent with the FFLO phase obtained from rf-penetration depth measurements [30] for organic sample $\kappa$-(BEDT-TTF)$_2$Cu(NCS)$_2$; (iv) NMR relaxation rate evidence for an additional phase transition line [31] and the clear upturn beyond the Pauli limit in the magnetic-field and angular-dependent high-resolution specific-heat measurements for the organic materials $\kappa$-(BEDT-TTF)$_2$Cu(NCS)$_2$ and $\beta''$-(ET)$_2$SF$_5$CH$_2$CF$_2$SO$_3$ [32,33], (v) magnetic torque evidence for the tricritical point between the FFLO, homogeneous superconducting, and paramagnetic metallic phases in the 2D magnetic-field-induced organic superconductor $\lambda$-(BETS)$_2$FeCl$_4$ [34]; (*vi*) measurements of the temperature and angular dependencies of $H_{c2}$ in pnictide superconductor LiFeAs [35] and KFe$_2$As$_2$ [36] as well as a dip in the interlayer resistance in parallel fields observed in the compound $\lambda$-(BETS)$_2$Fe$_{1-x}$Ga$_x$Cl$_4$ for $x = 0.37$ [37] (for all types of salts, ($x = 1.0$, $x = 0.37$, $x = 0$) signs for the FFLO state have been observed [23,38]) probably suggest that these systems may have realized the FFLO phase. Very recently, the first magnetocaloric and calorimetric observations of a magnetic-field-induced first order phase transition within a superconducting state to the paramagnetic state with higher entropy in $\kappa$-(BEDT-TTF)$_2$Cu(NCS)$_2$ molecular compound [39] has been reported. These results provide a strong thermodynamic evidence of the FFLO superconducting phase.

Similarly, quasi-1D superconducting compounds have been studied extensively in order to find out if this class of materials can exhibit the inhomogeneous state. The field-amplitude and field-angle dependence of the superconducting transition temperature $T_c(H)$ of the organic superconductor (TMTSF)$_2$ClO$_4$ in magnetic field applied along the conduction planes have been reported [40,41]. The authors observed an upturn of the curve of the upper critical field at low temperatures (An enhancement of almost two times over $H_P \simeq 27$ kOe is observed, $H_{c2} \simeq 50$ kOe [40]). Moreover, an unusual in-plane anisotropy of $H_{C2}$ in the high-field regime was observed. Both observations were interpreted as an evidence of the FFLO state formation [42–44].

The search for FFLO has also covered other low-dimensional conductors [45]. In particular, a possible realization of the FFLO phase in ultrathin crystalline Al films has been recently reported [46]. In superconductor-ferromagnet (S/F) bilayers, a quasi-one-dimensional FFLO-like state can be realized by Cooper pairs migrating from the superconductor into the ferromagnet. Due to the exchange splitting in the ferromagnet, the Cooper pair gains a nonzero momentum, resulting in an oscillating pairing wave function [47]. The observation of a $\pi$-junction predicted in [48] can be considered as the first direct evidence of the FFLO-like state [49,50].

Besides the quasi-low dimensional organic materials, heavy-fermion systems were shown to be favorable to the formation of the spatially modulated state [51–53]. The heavy fermion superconductor, CeCoIn$_5$, appears to meet all the strict requirements placed on the existence of the FFLO state. In a strong magnetic field, the superconducting transition is of the first order at low temperatures for both directions of the magnetic field. The first order transition at the upper critical field indicates that the Pauli paramagnetic effect dominates over the orbital effect for both field directions. A second anomaly is the heat capacity that has been observed within the superconducting phase. However, the phase found between the normal metal and the homogeneous superconductor show features expected rather from the antiferromagnetically ordered phase than from FFLO (for example, thermodynamic experiments showed an entropy decrease while shifting from the homogeneous phase to that one [54]). Nevertheless, recent measurements revealed an additional order intertwined with another one, probably an FFLO state [55,56].

Besides condensed matter physics, the FFLO phase has been predicted to occur also in other systems, such as cold atomic gases [57], quantum chromodynamics, nuclear physics, and astrophysics [58,59].

Various theoretical studies of the FFLO phase were performed in order to provide unambiguous ways to detect the manifestation of the FFLO phase. For instance, a tool based on the Josephson

effect was suggested, since the application of the magnetic fields to a junction recovers Josephson current that is suppressed between a conventional and the FFLO superconductors [60]. FFLO states in multiband superconductors have been analyzed as well. The analysis shows that the $s^{\pm}$ multiband pairing facilitates the FFLO transition and results in the upward curvature of the upper critical field in the intermediate temperature range [61]. The multiband $s^{\pm}$ pairing with several "shallow" bands with bottoms close to the FS can result in the FFLO state caused by a small shift of the chemical potential. This may enable one to tune the FFLO transition. In [62], it has been shown that the FFLO state is most stable when the Fermi level is in the proximity of one of the system energy subbands.

In this work, we provide an overview of our recent investigations on possible tools for exhibiting the FFLO phase based on its main feature, namely, the spatial modulations of the order parameter.

## 2. General Settings

In this section, we consider quasi-2D superconducting materials and model a layered superconductor as a stack of conducting layers (see Figure 2). The single-electron spectrum of such system is taken as follows

$$\zeta_{\mathbf{p}} = \frac{p_x^2}{2m_x} + \frac{p_y^2}{2m_y} + 2t \cos\left(p_z d\right) - \mu, \tag{1}$$

where the effective mass approximation is used for description of the in-plane charge carriers motion and the tight-binding approximation for the motion along the $z$-direction, perpendicular to the conducting layers. The Fermi surface acquires the shape of a corrugated cylinder with an elliptical cross section (see Figure 3). We assume that the corrugation of the Fermi surface, because of the coupling between layers, is small, i.e., $t \ll T_{c0}$, but sufficiently large to make the mean field treatment justified, namely the coupling between layers, $t$, satisfies the condition $T_{c0}^2 / E_F \ll t \ll T_{c0}$ [63,64]. Here, $T_{c0}$ is the critical temperature of the system at $H = 0$. In this model, an effective mass anisotropy can be taken into account by the scaling transformation of the coordinates and orbital magnetic field [65].

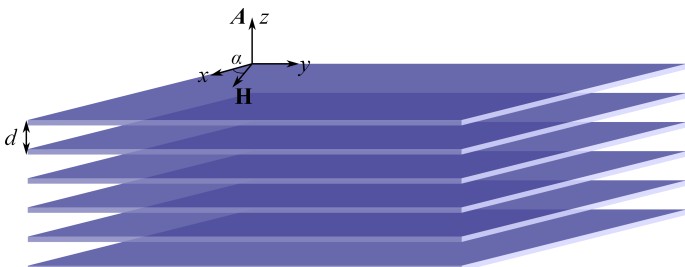

**Figure 2.** Scheme of the quasi-2D layered metal.

We choose the magnetic field to be parallel to the conducting planes and with a gauge for which the vector potential $\mathbf{A} = \mathbf{H} \times \mathbf{r}$ ($\mathbf{r} = (x, y, 0)$ is a coordinate in $xy$-plane), i.e., $A_z = -xH \sin\alpha + yH \cos\alpha$, where $\alpha$ is the angle between the applied field, with amplitude $H$, and $x$-axis. Assuming that the vector potential varies slowly at the interlayer distances and taking into account that the system is near the second-order phase transition, we can employ the linearized Eilenberger equation for a layered superconductor in the presence of the parallel magnetic field. It is the momentum representation with respect to the coordinate $z$ [1,66,67]:

$$\left(\Omega_n + \frac{1}{2}\widehat{\Pi}\right) f_\omega\left(\mathbf{n}, \mathbf{r}, p_z, k_z\right) = \left\{\Delta_{\widehat{\mathbf{p}}}(\mathbf{r}, k_z)\right.$$
$$\left. + \frac{\langle f_\omega\left(\mathbf{n}, \mathbf{r}, p_z, k_z\right)\rangle}{2\tau}\right\} \text{sign}(\omega_n). \tag{2}$$

Here, we use the notation

$$\widehat{\Pi} \equiv \hbar \mathbf{v}_F \cdot \nabla + 4it \sin(p_z d) \sin(\mathbf{Q} \cdot \mathbf{r} - \frac{k_z}{2} d), \tag{3}$$

where $\mathbf{Q} = (\pi dH/\phi_0)[-\sin\alpha, (m_x/m_y)^{1/2}\cos\alpha, 0]$ with $\phi_0 = \pi\hbar c/e$, $h = \mu_B H$ is the Zeeman energy, $\mathbf{v}_F = v_F\widehat{\mathbf{p}}$ is the in-plane Fermi velocity, $\tau$ is the impurity scattering time, and $\Omega_n \equiv \omega_n - ih + \text{sign}(\omega_n)/2\tau$. In this work, we consider both *s*-wave and *d*-wave pairing interaction. For the case of *d*-wave pairing, the interaction may be represented in the form

$$V_d\left(\mathbf{p}, \mathbf{p}'\right) = -\lambda\, 2\gamma\left(\widehat{\mathbf{p}}\right)\gamma\left(\widehat{\mathbf{p}}'\right), \tag{4}$$

where $\gamma_{d_{x^2-y^2}}\left(\widehat{\mathbf{p}}\right) = \widehat{p}_x^2 - \widehat{p}_y^2 = \cos\left(2\varphi\right)$ with $\varphi$, the azimuthal angle of $\mathbf{p}$ (angle between the momentum in the crystalline *xy*-plane and *x*-axis). In this expression, factor 2 is introduced for normalization. Then, the order parameter may be written as

$$\Delta_{\widehat{\mathbf{p}}}\left(\mathbf{r}\right) = \Delta_d\left(\mathbf{r}\right)\gamma\left(\widehat{\mathbf{p}}\right), \tag{5}$$

where $\Delta_d\left(\mathbf{r}\right)$ is defined as self-consistency relation,

$$\frac{1}{\lambda}\Delta_d\left(\mathbf{r}, k_z\right) = 2\pi T\Re \sum_{\omega>0}\left\langle \gamma\left(\widehat{\mathbf{p}}\right) f_\omega\left(\widehat{\mathbf{p}}, \mathbf{r}, p_z, k_z\right)\right\rangle, \tag{6}$$

where $\lambda$ is the pairing constant and the brackets denote averaging over $p_z$ and $\widehat{\mathbf{p}} = \mathbf{n}$,

$$\left\langle ...\right\rangle \equiv \int\limits_{-\frac{\pi}{d}}^{\frac{\pi}{d}} \frac{d\,dp_z}{2\pi} \int\limits_0^{2\pi} \frac{d\varphi}{2\pi}\left(...\right). \tag{7}$$

We assume that the temperature unit is so chosen that the Boltzmann constant $k_B = 1$. For the case of an *s*-wave pairing interaction, $\gamma\left(\widehat{\mathbf{p}}\right) = 1$.

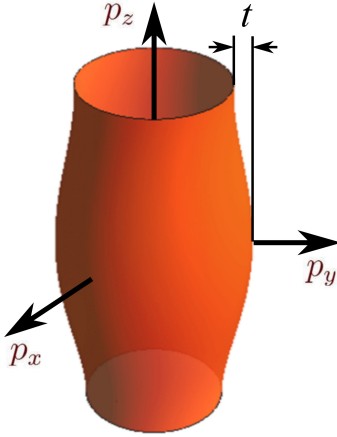

**Figure 3.** The Fermi surface of the layered metal in the form of a corrugated cylinder.

The FFLO state appears as a phase with a modulated order parameter with a wave vector $\mathbf{q}$ whose direction is determined by the crystal field effects [68–70] and the pairing symmetry [71]. The orientation of the FFLO modulation vector is arbitrary in a pure Pauli limited *s*-wave superconductor in the case of a Fermi surface with an elliptical cross section, which can be mapped by scaling transformation to an isotropic case [65]. For a *d*-wave superconductor, the symmetry of

superconducting pairing fixes the directions of the FFLO modulation vector even in the absence of the crystal field effects [71]. If the orbital effects are essential, the actual direction will be determined by the interplay between the anisotropy of pairing and the crystal field effects. For the triclinic symmetry of the organic superconductors, the FFLO modulation is pinned in a certain direction. The magnitude and the direction of the FFLO modulation vector are determined by the condition of the maximum value of the critical field

$$H_{cP2} = \max H_{c2}(q, \phi).$$

The simple exponential solution $f_\omega(\mathbf{n_p}, \mathbf{r}, p_z) \sim \exp(i\mathbf{qr})$ is no more valid in quasi-2D superconductors. Therefore, to describe correctly the angular dependence of the upper critical field in the FFLO phase, we have to account for the orbital effects, which add the higher harmonics in FFLO modulation, $\mathbf{q} \pm m\mathbf{Q}$. The solution of Equation (2) might be written as

$$f_\omega(\widehat{\mathbf{p}}, \mathbf{r}, p_z) = e^{i\mathbf{qr}} \sum_m e^{im\mathbf{Qr}} f_m(\omega_n, \widehat{\mathbf{p}}, p_z). \tag{8}$$

Because of the form for $f_\omega(\widehat{\mathbf{p}}, \mathbf{r}, p_z)$ in Equation (8), one can write the order parameter $\Delta(\mathbf{r})$ as

$$\Delta(\mathbf{r}) = e^{i\mathbf{qr}} \sum_m e^{i2m\mathbf{Qr}} \Delta_{2m}. \tag{9}$$

From symmetry considerations, it follows that $\Delta_{-2m} = \Delta_{2m}$. Substituting Equations (8) and (9) into Equation (2), one gets the following system of coupled equations [72]:

$$L_n(\mathbf{q}) f_0 + \widetilde{t} f_{-1} - \widetilde{t} f_1 = \Delta_0, \tag{10}$$
$$L_n(\mathbf{q} \pm \mathbf{Q}) f_{\pm 1} \pm \widetilde{t} f_0 \mp \widetilde{t} f_{\pm 2} = 0, \tag{11}$$
$$L_n(\mathbf{q} \pm 2\mathbf{Q}) f_{\pm 2} \pm \widetilde{t} f_{\pm 1} \mp \widetilde{t} f_{\pm 3} = \Delta_{\pm 2}, \tag{12}$$
$$L_n(\mathbf{q} \pm 3\mathbf{Q}) f_{\pm 3} \pm \widetilde{t} f_{\pm 2} = 0, \tag{13}$$

where $L_n(\mathbf{q}) = \Omega_n + i v_F \mathbf{q}/2$ and $\widetilde{t} = t \sin(p_z d)$. Here, we have taken into account that $\Delta_{\pm(2m+1)} = 0$ and introduced the notation $f_m \equiv f_m(\omega_n, \mathbf{n_p}, p_z)$. This hierarchy of coupled equations is broken on the level of $f_{\pm 3}$ in order to obtain symmetric equations for the first two harmonics of the order parameter up to the second order with respect to the small parameter $t/T_{c0}$.

## 3. Angular Dependence of the Upper Critical Field

From Equation (10) we can obtain function $H_{c2}(q, \phi)$ that should be minimized in order to obtain the absolute value and the direction of the modulation vector. In the Pauli limit, when neglecting the orbital motion, we obtain

$$\ln\left(\frac{T_{c0}}{T_{cP}}\right) = \pi T_{cP} \sum_n \frac{1}{\omega_n} - \left\langle \frac{2\gamma^2(\widehat{\mathbf{p}})}{L_n(\mathbf{q})} \right\rangle \equiv F(\tilde{h}_{cP}, \tilde{q}_{cP}), \tag{14}$$

where the renormalized variables $\tilde{h} = h/2\pi T$ and $\tilde{q} = q/2\pi T$ are introduced,

$$F(\tilde{h}, \tilde{q}) \equiv \pi T \sum_n \frac{1}{\omega_n} - \frac{2}{g_1^v} - \frac{2\left(g_1^v - 2\Omega_n\right)^2}{g_1^v \left(g_1^v + 2\Omega_n\right)^2} \cos(4\phi). \tag{15}$$

$T_{cP}$ is the temperature of the onset of the superconductivity in the pure Pauli regime, $g_1^v \equiv \sqrt{q^2 v_F^2 + 4\Omega_n^2}$, and $\phi$ is the angle the $\mathbf{q}$ vector makes from the x-axis. The $H_{cP2}$ and the modulation

vector, maximizing $H_{cP2}$, are illustrated in Figure 1. In the case of an *s*-wave superconductor (obtained from Equation (15) by substitution $\phi = \pi/8$)

$$F(\tilde{h}, \tilde{q}) \equiv \pi T \sum_n \frac{1}{\omega_n} - \frac{2}{g_1^v}. \tag{16}$$

For a *d*-wave superconductor, the paramagnetic upper critical field is never smaller than $H_{cP2}$ for an *s*-wave superconductor [71], as shown in Figure 4. The optimal direction of the modulation vector in a *d*-wave superconductor is $\phi = \pm\pi/4, \pm3\pi/4$ for $T^{**} < T < T^*$ and $\phi = 0, \pm\pi/2, \pi$ for $0 < T < T^{**}$ with $T^{**} \simeq 0.056 T_{c0}$. $H_{cP2}$ and $\mathbf{q}$ in *s*-wave pairing symmetry is shown in the Figure 4a.

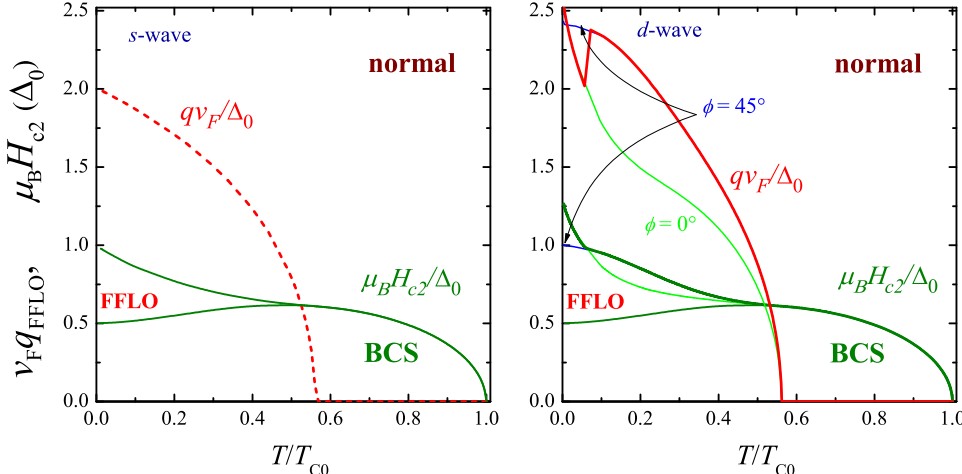

**Figure 4.** The upper critical field $H_{cP2}$ and the absolute value of the FFLO modulation vector $q$ as a function of $T_{c0}$ for *s*-wave (left panel) and *d*-wave (right panel) superconductors in the pure Pauli limit.

Let us now account for the orbital effects. As it has been seen above [73], in the absence of orbital effect, the FFLO state only appears at $T < T^* \simeq 0.56 T_{c0}$ and $H > H^* \simeq 1.06 T_{c0}/\mu_B$. Therefore, the order of the magnitude of the magnetic field required to observe the FFLO state can be found from the relation $\mu_B H \sim T_{c0}$. Taking this into account, one obtains $v_F Q \sim v_F e d T_{c0}/\mu_B c \sim (d/a) T_{c0}$, where $a$ is the unit cell in the *xy*-plane. Therefore, $v_F Q \gtrsim T_{c0}$. Due to the assumption $t \ll T_{c0} \lesssim v_F Q$, one has $\sqrt{t T_{c0}} \ll v_F Q$. This condition allows us to hold only the terms up to the first harmonics in Equations (8) and (9). Then, one simply gets

$$L_n(\mathbf{q}) f_0 + \tilde{t} f_{-1} - \tilde{t} f_1 = \Delta_0, \tag{17}$$

$$L_n(\mathbf{q} \pm \mathbf{Q}) f_{\pm 1} \pm \tilde{t} f_0 = 0. \tag{18}$$

The solution of this system of equations is

$$\frac{T_{cP} - T_c}{T_c} \equiv aAt^2, \tag{19}$$

where the following notations are used

$$a = \pi T \sum_{n,\xi=\pm} T_n(\mathbf{q}, \mathbf{q}, \xi\mathbf{Q})|_{T=T_{cP}}, \tag{20}$$

$$T_n(\mathbf{g}_1, \mathbf{g}_2, \mathbf{g}_3) = \frac{1}{2} \int_0^{2\pi} \frac{d\varphi}{2\pi} \frac{2\cos^2(2\varphi)}{L_n(\mathbf{g}_1) L_n(\mathbf{g}_2) L_n(\mathbf{g}_3)}, \tag{21}$$

$$A \equiv 1 - \frac{h}{T_{cP}} \frac{\partial T_{cP}}{\partial h} = \frac{1}{1 - \frac{\hbar \partial F(\tilde{h}, \tilde{q})}{\partial \tilde{h}}\Big|_{T = T_{cP}}}. \tag{22}$$

Here, the expression for $T_n(\mathbf{g}_1, \mathbf{g}_2, \mathbf{g}_3)$ can acquire an analytical form

$$T_n(\mathbf{g}_1, \mathbf{g}_2, \mathbf{g}_3) = -8t^2 \sum_{k=1}^{N_k} \frac{4a}{p_k^2 p_{k+1} p_{k+2}}$$

$$+ \frac{ig_k^2}{\left(g_k^- - g_k^+\right)\left(g_k^- - g_{k+1}^-\right)\left(g_k^- - g_{k+1}^+\right)}$$

$$\times \frac{\left[1 + \left(g_k^-\right)^4\right]}{\left(g_k^- - g_{k+2}^-\right)\left(g_k^- - g_{k+2}^+\right)\left(2a - g_k^v\right)}, \tag{23}$$

where we have introduced the following notations $g_k^v \equiv \sqrt{\mathbf{g}_k^2 v_F^2 + 4\Omega_n^2}$, $g_k = g_{k,x} - ig_{k,y}$, $\mathbf{g}_1 \equiv \mathbf{q}$, $\mathbf{g}_2 \equiv \mathbf{q} \pm \mathbf{Q}$, $\mathbf{g}_3 \equiv \mathbf{q} \pm 2\mathbf{Q}$ and

$$g_k^\pm \equiv i \frac{\left(2\Omega_n \pm g_k^v\right)}{g_k v_F}, \tag{24}$$

where $k$ is the cycling index with $k = 1, 2, 3$.

In our numerical investigations, we restrict ourselves to the parameters of the compound $\kappa$-(BEDT-TTF)$_2$Cu(NCS)$_2$. The Maki parameter, $\alpha \simeq 8$ [24], the interlayer coupling is $t = 1.5$ K (1.8 K) [74], $t/T_{c0} = 0.16$ (0.2), $\Delta_0 = 2.8kT_{c0}$ [75] and the Fermi velocity $v_F = 5.0 - 10.0 \times 10^4$ m/s [76]. We have chosen the value $v_F = 7.5 \times 10^4$ m/s [74]. Introducing the dimensionless Fermi velocity parameter, $\eta = \hbar v_F \pi d / \phi_0 \mu_B$, this value of $v_F$ corresponds to $\eta = 2.55$. The interlayer distance is $d = 1.62$ nm. The summation over the Matsubara frequencies was performed numerically.

In the following for the case of a $d$-wave superconductor, we consider two situations. In the first one, the symmetry of superconducting pairing fixes the direction of the FFLO modulation wave vector, characterized by $\phi$, the angle that $\mathbf{q}$ makes from the $x$-axis: at high temperatures of the FFLO phase, $\phi = \pm\pi/4, \pm 3\pi/4$, while, in the low-temperature phase $\phi = 0, \pm\pi/2, \pi$ thus making it four-fold degenerate. In the second case, the crystal field effect leaves this degeneracy.

Figure 5 presents the orbital-motion induced normalized correction of $\Delta T_c = T_c - T_{cP}$ as a function of $T_{cP}/T_{c0}$ for different directions of the applied field when the symmetry of superconducting $d$-wave pairing fixes the direction of the FFLO modulation wave vector. The direction of the external field is measured by $\alpha$, the angle the applied field $\mathbf{H}$ makes from the $x$-axis. Due to the four-fold symmetry of the $d_{x^2-y^2}$ pairing, we plot $\Delta T_c(T_{cP}/T_{c0})$ curves only for $\alpha = 0, \pi/9, \pi/4$. The regions $T_{cP} < T^{**}$ and $T_{cP} > T^*$ are shadowed. In the low-temperature region, the modulation vector $\mathbf{q}$ is along the $x$-axis, while at $T_{cP} > T^* - |\mathbf{q}| = 0$, and it describes $\Delta T_c$ in the conventional phase. The central domain illustrates $\Delta T_c$, when the modulation wave vector $\mathbf{q}$ makes the angle $\phi = \pi/4$ with the $x$-axis. The re-orientational transition at $T_{cP} < T^{**}$ is accompanied by a strong change of the orbital correction, which should result in the sharp step on the $T_c(H)$ curve.

The existence of the nodes in the order parameter results in particular features of the anisotropy of the superconducting onset temperature induced by the spatially modulated FFLO phase (in the case of $d$-wave only, when the symmetry of superconducting pairing fixes the direction of the FFLO modulation wave vector). Figure 6 shows the magnetic field angular dependence of the normalized superconducting transition temperature, $T_c(\alpha)/T_{cP}$, calculated at $T_{cP}/T_{c0} \simeq 0.03$, 0.056, when the modulation vector is fixed by the symmetry of pairing to the direction of maxima of the order parameter, and at $T_{cP}/T_{c0} \simeq 0.057$, 0.075, 0.15 and 0.4, when it is fixed to the nodes of $\Delta_{\hat{\mathbf{p}}}(\mathbf{r})$. In the polar plot, the direction of each point seen from the origin corresponds to the magnetic field direction and the distance from the origin corresponds to the normalized critical temperature. One can see that the field-angle dependence of the onset of superconductivity makes evident two transitions: one at $T_{cP}/T_{c0} = 0.56$,

the transition from the conventional phase to the FFLO modulated phase, accompanied by a rotation of the anisotropy principal axis by the angle $\pi/4$. The second transition occurs at $T_{cP}/T_{c0} \simeq 0.056$, when the direction of the FFLO modulation vector rotates by $\phi = \pi/4$, which is reflected in a change of the overall anisotropy of the onset of superconductivity. The transition from the high temperature FFLO phase to the low temperature phase is accompanied by a strong decrease of the orbital effect that makes $T_c(\alpha)/T_{cP}$ dependence at $T < T^{**}$ much closer to the pure paramagnetic limit.

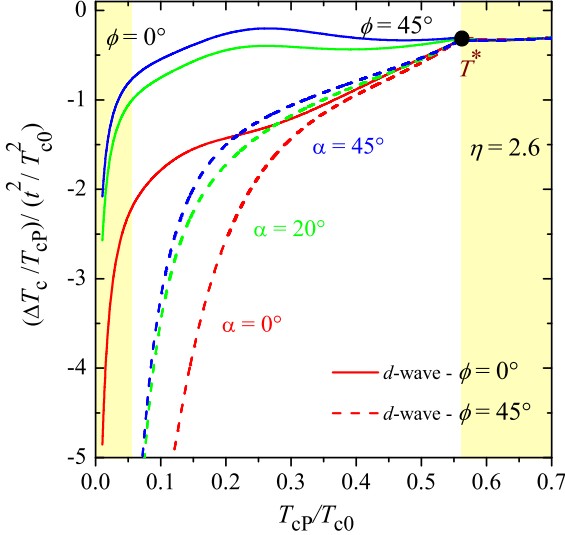

**Figure 5.** Contribution of the orbital effect as a function of $T_{cP}/T_{c0}$ for several field directions when $\eta = 2.6$. The regions showing the results obtained when **q** is along the *x*-axis and for the conventional phase are highlighted in yellow, while the white region is for the case when **q** vector makes angle $\phi = \pi/4$ with the *x*-axis.

In real systems, the crystal field effect is inevitably present and influences the pinning direction of the FFLO modulation [68,69]. As already mentioned previously, the orientation of the FFLO modulation vector is arbitrary in the pure Pauli limited regime for an elliptic Fermi surface (*s*-wave case). The crystal field introduces deviations from the ellipticity and pins the FFLO modulation vector in a certain direction. In the case of a *d*-wave superconductor and weak crystal effect, it can lift the four-fold degeneracy of the direction of FFLO modulation, making it two-fold. To illustrate this, we pin the FFLO modulation vector along the $\phi = \pi/4, 5\pi/4$ directions and investigate the anisotropy of the superconducting onset in the temperature range $T^{**} < T$. Figure 7 displays the orbital-motion induced normalized correction of the transition temperature, $\Delta T_c$, as a function of normalized temperature, $T_{cP}/T_{c0}$, for several orientations of the applied field (for both *s*-wave and *d*-wave cases). Here, the crystal field effect is sufficient to break the four-fold degeneracy to fix the direction of the FFLO modulation vector. The dashed lines illustrate the result for $\Delta_{\pm 2} = 0$, considered in this section, while the solid lines are the solutions with $\Delta_{\pm 2} \neq 0$, discussed in the next section.

As it was intuitively expected, the orbital effects reduce the superconducting onset temperature, $\Delta T_c < 0$ in both cases. While increasing the applied magnetic field, $\Delta T_c$ first decreases in most cases until the tricritical point, $H^*$, is reached [77]. At $H > H^*$, the function $\Delta T_c(H)$ strongly depends on the in-plane effective mass anisotropy (not considered here and investigated in [42] and angle $\alpha$ (and this is very different from the conventional phase). For some angles, $\Delta T_c$ may exhibit an upturn and $T_c$ may approach the paramagnetic limit, $T_{cP}$, when $H$ increases. In contrast, for small $\alpha$, an increase of the magnetic field leads to a decrease of $\Delta T_c$. For intermediate angles, $\Delta T_c$ can be a non-monotonic function of the field strength. One can infer that the strong field-direction dependence of the superconducting onset temperature, $T_c(\alpha)$, appears at high magnetic fields when the FFLO state develops, while it is absent at low fields.

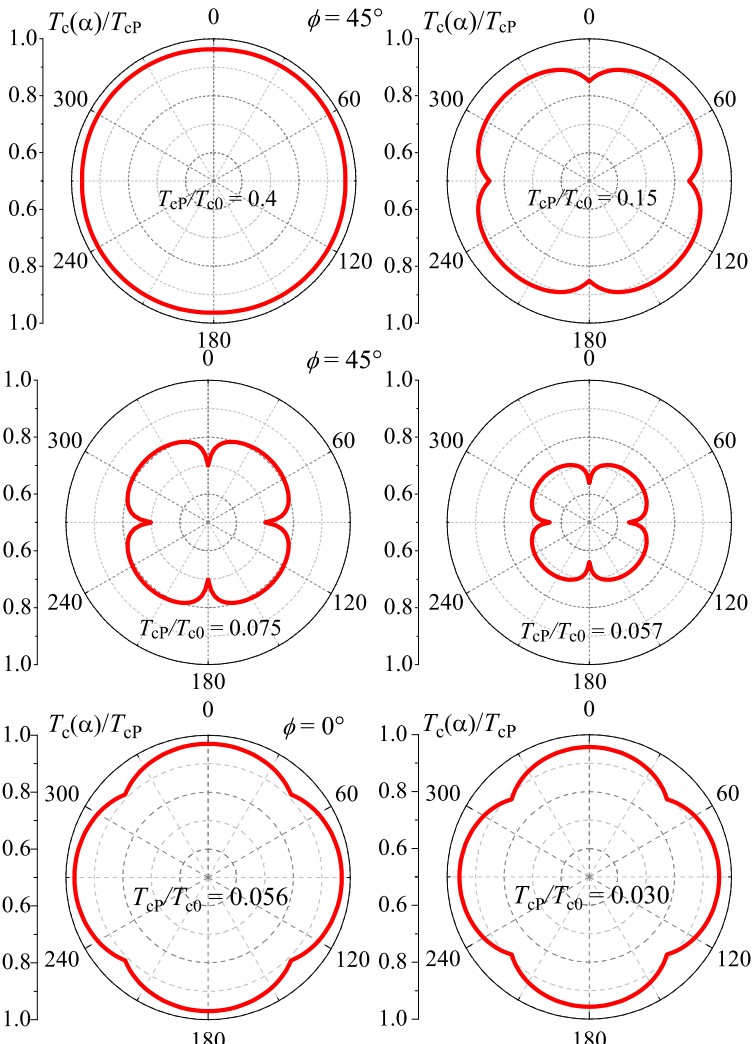

**Figure 6.** Normalized superconducting transition temperature, $T_c(\alpha)/T_{cP}$ as a function of the $\alpha$ for several values of $T_{cP}/T_{c0}$ and $\eta = 2.6$. The angle between the **q**-vector and $x$-axis is $\phi = \pi/4$ (first and second panels) and $\phi = 0$ (last panel). The direction of **q**-vector is fixed by the symmetry of the order parameter.

The change in the anisotropy of the superconducting onset temperature that is induced by the FFLO phase is particularly visible in Figures 8 and 9, where the same plot is given as in the Figure 6 but for the *s*-wave and *d*-wave cases. For magnetic fields below $H^*$, the behavior of the upper critical field is isotropic. When increasing $H$ above $H^*$, a strong in-plane anisotropy of $H_{c2}$ develops, which remains and becomes essentially pronounced at high fields. In particular, this behavior develops with external magnetic field for the case of the isotropic in-plane Fermi surface. The maximum transition temperature is for the magnetic field orientation perpendicular to the direction of the FFLO modulation vector along the $x$-axis. In Figure 9, we see that the shape of the field-angle dependence of the onset of superconductivity in the high temperature FFLO phase is similar to that obtained for an *s*-wave superconductor [42]. However, the principal axis of the plot is not vertical but tilted by $\pi/4$ and fixed along the direction of the modulation vector. Even in the presence of the "easy axis" along the direction $\pi/4$ induced by the crystal field, the four-fold degeneracy of the direction of the FFLO modulation vector restores again for $T < T^{**}$. Therefore, anisotropy in the low temperature FFLO phase is presumably the same as that discussed for Figure 6.

The results discussed in this section show that, in the FFLO phase, the in-plane upper critical field anisotropy in quasi-low-dimensional superconductors is settled by the interplay between the

modulation and magnetic field wave vectors. The superconducting onset temperature is maximal for the field oriented perpendicular to the FFLO modulation vector. The change of the anisotropy of the critical field as well as of its fine structure may give important information about the FFLO state and unambiguously prove its existence. Our calculations support the interpretation of the experimentally observed in-plane anisotropy of the onset of superconductivity in $(TMTSF)_2ClO_4$ samples as a realization of the FFLO state with the modulation vector close to the **b**\*-axis [40].

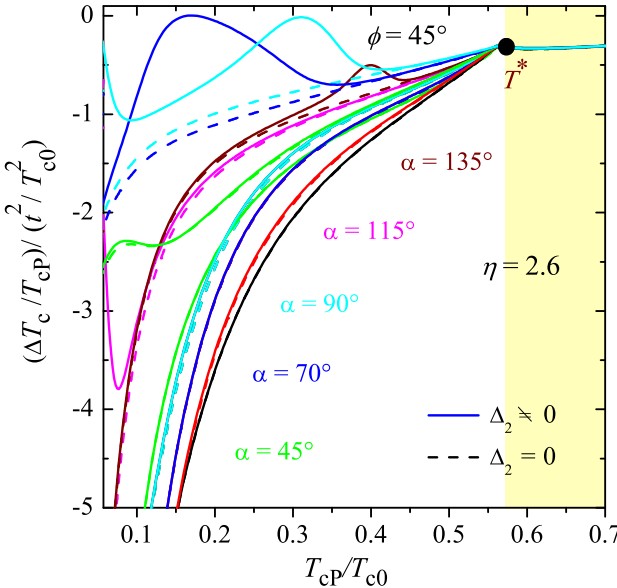

**Figure 7.** Contribution of the orbital effect as a function of $T_{cP}/T_{c0}$ for several field directions when $\eta = 2.6$. The case of d-wave with **q** vector making angle $\phi = \pi/4$ with the *x*-axis as well as the s-wave superconductor are shown.

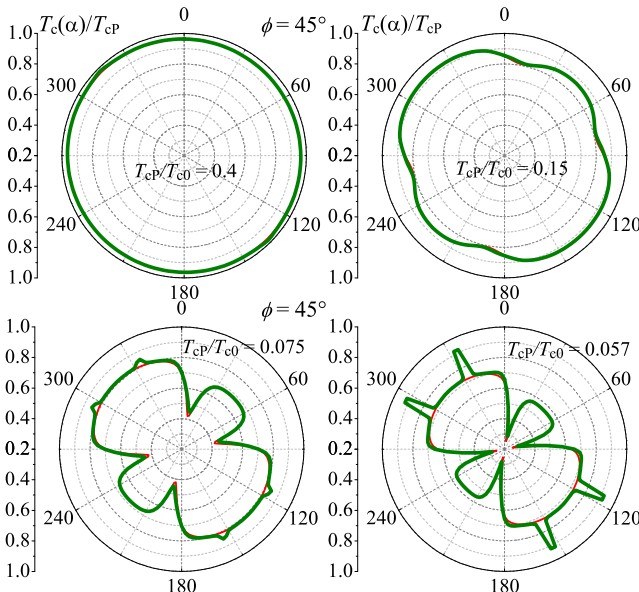

**Figure 8.** The same as in Figure 6; however, the direction of **q**-vector is influenced by the crystal field.

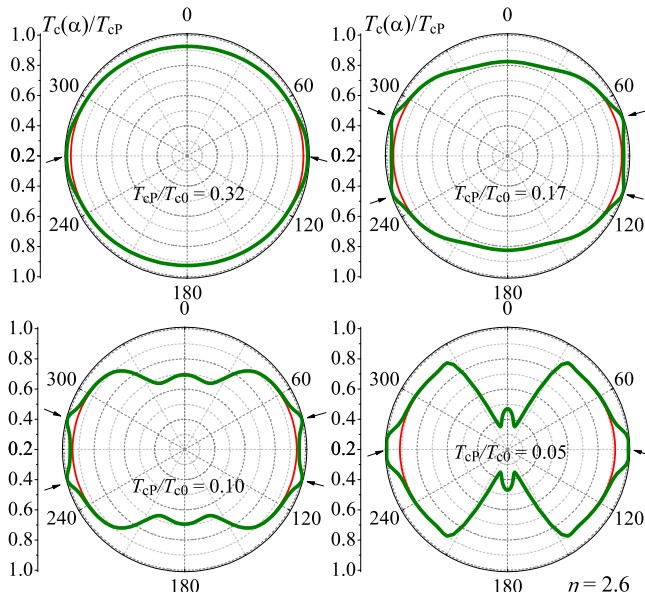

**Figure 9.** The same as in Figure 6; however, the direction of **q**-vector is influenced by the crystal field. Here is the *s*-wave case. Thick lines are for $\Delta_2 \neq 0$; thin lines are for $\Delta_2 = 0$.

## 4. Resonance In-Plane Magnetic Field Effect

We know that the vector potential of the parallel magnetic field results in a modulation of the interlayer coupling with the period $\lambda_H = 2\phi_0/dH$, where $d$ is the interlayer distance and $\phi_0 = \pi c/e$. In this section, we show that the period of this modulation may interfere with the in-plane FFLO modulation leading to the anomalies in the critical field behavior. The strongest effect occurs when the magnetic length $\lambda_H$ coincides, i.e., in the resonance with the period of FFLO modulation $\lambda_{\text{FFLO}}$ (for $T = 0$, $\lambda_{\text{FFLO}}^0 = \pi\hbar v_F/\Delta_0 = \pi^2\xi_0$). The upper critical field at $T = 0$ observed in $\kappa$-(BEDT-TTF)$_2$Cu(NCS)$_2$ is $H_{c2} \simeq 30$ T. Since in this compound the interlayer distance $d = 1.62$ nm [78], $\xi_0 = 7$–9 nm this leads to $\lambda_H = 85$ nm and $\lambda_{\text{FFLO}}^0 = 70$–90 nm. In (TMTSF)$_2$ClO$_4$ $H_{c2}(0) = 5$ T, $d = 1.31$ nm, $\xi_0 = 45$ nm, resulting in $\lambda_H = 630$ nm and $\lambda_{\text{FFLO}}^0 = 444$ nm. In $\lambda$-(BETS)$_2$FeO$_4$, the lower $H_{c2}(0) = 18$ T, $d = 1.85$ nm, $\xi_0 = 8.5$ nm, resulting in $\lambda_H = 130$ nm and $\lambda_{\text{FFLO}}^0 = 85$ nm. For these compounds $\lambda_{\text{FFLO}}^0 < \lambda_H^0$ at $T = 0$. The FFLO modulation appears only at $T < T^* \simeq 0.56T_{c0}$ with a wave vector **q** growing from $q = 0$ to $q_0 = q(T = 0) = 2\Delta_0/\hbar v_F$ with decreasing temperature. Therefore, if the condition $\lambda_{FFLO}^0 < \lambda_H^0$ is satisfied at $T = 0$, then, at some finite temperature $T$, the resonance condition, $\lambda_{\text{FFLO}}(T) = \lambda_H(T)$, should be realized. It corresponds to the situation when the strongly overlapping Josephson vortices form the rectangular lattice with its centers just above the nodes of the order parameter (previously, such a mechanism of the pinning of Josephson vortices by the nodes of FFLO modulation was suggested in [79] (see Figure 10) and observed in organic superconductor $\lambda$-(BETS)$_2$FeCl$_4$ [80]).

From the system of Equations (10) and (13), it is seen that, if $L_n(\mathbf{q}) = L_n(\mathbf{q} \pm 2\mathbf{Q})$, then the averaged Equations (10) and (12) for $\Delta_0$ and $\Delta_{\pm 2}$ show that $\Delta_{\pm 2}$ is of the same order as $\Delta_0$. To account for such degenerate or resonance situations, Equations (12) and (13) are included in our consideration. Making use of the self-consistency relation, one obtains for $T < T^*$ in a second-order approximation on the small parameter $t/T_{c0}$

$$\Delta_0 \left( \frac{1}{\lambda} - \pi T_c \sum_n \left\langle \frac{2\gamma^2(\hat{\mathbf{p}})}{L_n(\mathbf{q})} \right\rangle_{\hat{\mathbf{p}}} + t^2 a \right) = \Delta_{\pm 2} t^2 c_\pm, \tag{25}$$

$$\Delta_2 \left( \frac{1}{\lambda} - \pi T_c \sum_n \left\langle \frac{2\gamma^2(\hat{\mathbf{p}})}{L_n(\mathbf{q} \pm 2\mathbf{Q})} \right\rangle_{\hat{\mathbf{p}}} + t^2 b_\pm \right) = \Delta_0 t^2 c_\pm, \tag{26}$$

where, additionally to Equation (20), the following are introduced:

$$b_\pm = \pi T \sum_{n,\xi=\pm} T_n \left( \mathbf{q} \pm 2\mathbf{Q}, \mathbf{q} \pm 2\mathbf{Q}, \mathbf{q} \pm 2\mathbf{Q} + \xi \mathbf{Q} \right)\big|_{T=T_{cP}}, \tag{27}$$

$$c_\pm = \pi T \sum_{n} T_n \left( \mathbf{q}, \mathbf{q} \pm \mathbf{Q}, \mathbf{q} \pm 2\mathbf{Q} \right)\big|_{T=T_{cP}}. \tag{28}$$

Taking into account the fact that the critical temperature when accounting for the orbital effects, $T_c$, is close to $T_{cP}$, Equations (25) and (26) can be written as the following system of coupled equations:

$$\Delta_0 \left( -\frac{(T_{cP} - T_c)}{A T_c} + t^2 a \right) = \Delta_{\pm 2} t^2 c_\pm, \tag{29}$$

$$\Delta_{\pm 2} \left( -\frac{(T_{cP} - T_c)}{T_c B_\pm} + t^2 b_\pm + \delta_\pm \right) = \Delta_0 t^2 c_\pm, \tag{30}$$

where

$$B_\pm \equiv \frac{A \left. \frac{\hbar \partial \Lambda_\pm(\tilde{h})}{\partial \tilde{h}} \right|_{T=T_{cP}}}{A - \left. \frac{\hbar \partial \Lambda_\pm(\tilde{h})}{\partial \tilde{h}} \right|_{T=T_{cP}}}, \tag{31}$$

$$\Lambda_\pm = \pi T \sum_{n} \left\langle \frac{2\gamma^2(\hat{\mathbf{p}})}{L_n(\mathbf{q})} \right\rangle_{\hat{\mathbf{p}}} - \left\langle \frac{2\gamma^2(\hat{\mathbf{p}})}{L_n(\mathbf{q} \pm 2\mathbf{Q})} \right\rangle_{\hat{\mathbf{p}}}, \tag{32}$$

$$\delta_\pm = \Lambda_\pm\big|_{T=T_{cP}}. \tag{33}$$

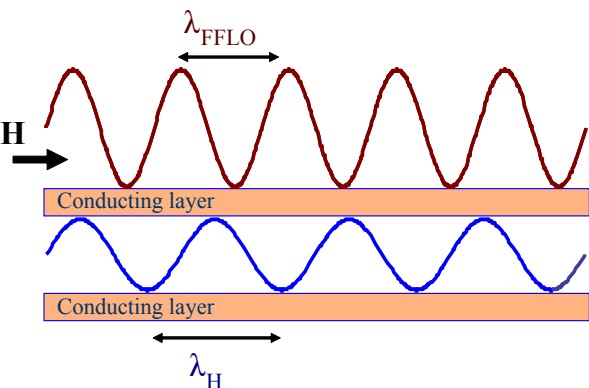

**Figure 10.** Schematic picture of the FFLO modulated order parameter and the oscillations of the interlayer coupling, explaining the physical origin of the resonances.

The solution of the system of Equations (29) and (30) is given as

$$\frac{T_{cP} - T_c}{T_c} \equiv \frac{(aA + b_\pm B_\pm)\, t^2 + B_\pm \delta_\pm}{2}$$

$$+ \frac{1}{2} \left\{ \left[ (aA + b_\pm B_\pm)\, t^2 + B_\pm \delta_\pm \right]^2 \right.$$

$$\left. - 4AB_\pm \left[ \left( ab_\pm - c_\pm^2 \right) t^4 + at^2 \delta_\pm \right] \right\}^{\frac{1}{2}}, \tag{34}$$

where, for $\pm$, the values are chosen that maximize the critical temperature. Usually, the second harmonic of the order parameter, $\Delta_{\pm 2}$, can be neglected because the developed theory is valid up to the second order with respect to $t/T_{c0}$, and because $t \ll T_{c0} \lesssim v_F Q$. In this case, the solution of the

system is just $(T_{cP} - T_c)/AT_c = -t^2 a$, as it was shown in the previous section. However, if $\delta_{\pm} = 0$ (the resonance condition), the term $-(T_{cP} - T_c)/T_c B_{\pm} + t^2 b_{\pm} + \delta_{\pm}$ in the l.h.s. of Equation (30) is of the same order as the corresponding term in Equation (29) (up to the second order with respect to $t/T_{c0}$). Consequently, $\Delta_{\pm 2}$ becomes on the order of $\Delta_0$, and one has to consider both equations in the system on equal footing [81].

Figure 7 displays the change of the transition temperature, $\Delta T_c = T_c - T_{cP}$, due to the orbital effects of the applied magnetic field as a function of the reduced temperature for different field orientations. The solid curves display the results of the second harmonic correction, and they are the solutions of the full system of these equations. One can see (more pronounced in the *s*-wave case) that two types of curves (dashed and solid) almost coincide for angles $\alpha = 0°, 20°, 45°$. However, there are essential differences in the curves' behavior for the angles $\alpha = 70°, 90°$. These differences are induced by the just discussed resonance effect. The curve $\Delta T_{cP}$ for $\alpha = 70°$ exhibits only one cusp at $T_{cP}/T_{c0} \approx 0.2$. $T_{cP}/T_{c0} \approx 0.2$ $\delta_{\xi}$ is close to zero and this vicinity induces a wide pseudo-resonance peak on the curve $\Delta T_{cP}$ for $\alpha = 70°$. Our results show that the resonance contribution can become on the order of the "orbital" effect itself. At exact resonances, the suppression of the critical temperature due to the orbital effect becomes negligible. The same conclusion is valid for the case of a *d*-wave superconductor.

The resonance effect results in particular features of the anisotropy of the superconducting onset temperature induced by the spatially modulated FFLO phase. The thick curves in Figures 8 and 9 show the magnetic field angular dependence of the normalized superconducting transition temperature, $T_c(\alpha)/T_{cP}$, when accounting for the second harmonics of the order parameter, $\Delta_{\pm 2} \neq 0$. We see that, in addition to the overall anisotropy induced by the FFLO modulation and studied in [42], cusps develop for certain directions of the applied field, when the resonance conditions are realized. These cusps are the result of the resonant interplay between the FFLO wave vector and the magnetic wave vector, when the orbital effects of the field are taken in the second-order approximation.

In this section, we have shown that, in layered superconductors under the applied in-plane magnetic field in the FFLO phase, the resonance between the modulation wave vector and the vector potential may lead to anomalous cusps in the field-direction dependence of the upper critical field. Therefore, we suggest that observation of characteristic cusps in the anisotropy of the onset of superconductivity may be direct evidence for the appearance of the FFLO phase in layered superconductors.

## 5. FFLO Lock-In Effect

In this section, we keep studying the effects of orbital contribution on the FFLO modulation, targeting unambiguous proof of the FFLO phase. We demonstrate the locking phenomena—the result of the interplay between two competing length scales: the period of the magnetic field potential and the characteristic length associated with the FFLO modulation [82]. The locking occurs when the modulation wave vector remains equal to the period (or matches *m* periods) of the potential through some range of the magnetic field values. To study effects of competing periodicities in the system, we use the Frenkel–Kontorova (FK) model [83,84].

In the rest part of the article for the purpose of simplicity, we consider a quasi-1D conductor with the following electron spectrum $E_{\mathbf{p}} = p_x^2/2m_x + 2t_y \cos(p_y d_y) + 2t_z \cos(p_z d_z)$, where $d_y$ and $d_z$ are the inter-chain distances along the *y*- and *z*-axis, respectively. We assume that the couplings between chains are small, i.e., $t_z < T_{c0}$ and $t_y < T_{c0}$, but sufficiently large to stabilize the superconducting long-range order and to make the mean field treatment justified, $T_{c0}^2/E_F \ll t_z$, $T_{c0}^2/E_F \ll t_y$ [63]. Here, $T_{c0}$ is the critical temperature of the system at $H = 0$. In quasi-1D superconductors, the orbital effect is extremely weak for the magnetic field applied along the chains. Indeed, near $T_{c0}$, the parallel (along the *x*-axis) upper critical field is $H_{c2}^x = \frac{\phi_0}{2\pi\xi_y\xi_z}\frac{T_{c0}-T}{T_{c0}}$, where $\xi_y = d_y\frac{t_y}{T_{c0}}$ and $\xi_z = d_z\frac{t_z}{T_{c0}}$, while the perpendicular upper critical field (i.e., along the *y*-axis) is $H_{c2}^y = \frac{\phi_0}{2\pi\xi_x\xi_z}\frac{T_{c0}-T}{T_{c0}}$, with $\xi_x \sim \frac{v_F}{T_{c0}}$. We see that $H_{c2}^x \gg H_{c2}^y$ and the orbital effect for the parallel component of the magnetic field is weakened by

the factor $\frac{t_y}{T_{c0}}\frac{d_y}{\xi_x} \ll 1$. Therefore, in our analysis of the orbital effect, it is enough to take into account only the perpendicular component of the magnetic field because it provides the dominant contribution to the orbital effect. For the purpose of clarity, we consider the magnetic field to be aligned in the $xy$-plane, making the angle $\theta$ with the $x$-axis. The perpendicular component of the magnetic field $H_y = H\sin\theta$, and the corresponding vector potential may be chosen as $A_z = -xH_y = -xH\sin\theta$, $A_y = 0$, $A_x = 0$. Then, the Eilenberger equations reduces to the following linearized Ginzburg–Landau equation (for details, see the Appendix)

$$
\begin{aligned}
\alpha\Delta\left(x\right) = {} & \beta\partial_x^2\Delta\left(x\right) - \delta\partial_x^4\Delta\left(x\right) - 48t^4\nu\sin^4\left(\chi_A\right)\Delta\left(x\right) \\
& - \frac{t^2}{2}\mu\left[1 - 7\cos\left(2\chi_A\right)\right]\left[\partial_x\chi_A\right]^2\Delta\left(x\right) \\
& + 2t^2\mu\left[3\sin\left(2\chi_A\right)\partial_x\chi_A\partial_x\Delta\left(x\right)\right. \\
& \left. + 3\sin^2\left(\chi_A\right)\partial_x^2\Delta\left(x\right)\right] - 8t^2\gamma\sin^2\left(\chi_A\right)\Delta\left(x\right),
\end{aligned}
\tag{35}
$$

where $Q = -\pi d_z H_y/\phi_0 = -\pi d_z H\sin\theta/\phi_0$ with $\phi_0 = \pi\hbar c/e$, $t \equiv t_z$, $\alpha \equiv \ln\frac{T_c}{T_{c0}} - \left[K_1\left(T_c\right) - K_1^0\left(T_c\right)\right] = \frac{T_c - T_{cP}}{AT_c}$ and $\alpha < 0$ in the uniform superconducting state. This equation contains the magnetic-field induced potential, which is periodic in real space with the period $\lambda_H = 2\phi_0/d_z H_y$. Equation (35) leads to the functional obtained in the limit of $t < v_F Q$ as [85]

$$
\begin{aligned}
F_{sn} = {} & \frac{1}{L_x}\int \mathrm{d}x\left\{\alpha\left|\Delta\left(x\right)\right|^2 + \beta\left|\partial_x\Delta\left(x\right)\right|^2 + \delta\left|\partial_x^2\Delta\left(x\right)\right|^2\right. \\
& + \frac{1}{2}t^2\mu\left[1 - 7\cos\left(2Qx\right)\right]Q^2\left|\Delta\left(x\right)\right|^2 \\
& + 6t^2\mu\sin^2\left(Qx\right)\left|\partial_x\Delta\left(x\right)\right|^2 \\
& \left. + 4t^2\gamma\left[1 - \cos\left(2Qx\right)\right]\left|\Delta\left(x\right)\right|^2\right\}.
\end{aligned}
\tag{36}
$$

The coefficients found in this expression are provided in the Appendix [86].

It is known [87] that, without the orbital contribution in 1D, the solution of the Ginzburg–Landau (GL) equation is the Jacobi elliptic sine function $\Delta\left(x\right) = \frac{v_F}{\xi(k)}k\,\mathrm{sn}\left(\frac{x}{\xi(k)},k\right)$, expressed in terms of the modulus $k$, which is determined upon minimizing the free energy at the fixed external field parameter $h$. This solution in the vicinity of the phase transition line takes the simple form, $\Delta_0\cos(qx)$ with $q$ the absolute value of the modulation wave vector. The linearized Ginzburg–Landau functional (36) and Equation (35) in the paramagnetic limit (when neglecting the orbital contribution) reduce to

$$
F_{sn}^{PM} = \frac{1}{L_x}\int \mathrm{d}x\left\{\alpha\left|\Delta\left(x\right)\right|^2 + \beta\left|\partial_x\Delta\left(x\right)\right|^2 + \delta\left|\partial_x^2\Delta\left(x\right)\right|^2\right\},
\tag{37}
$$

and $\alpha\Delta\left(x\right) = \beta\partial_x^2\Delta\left(x\right) - \delta\partial_x^4\Delta\left(x\right)$, respectively, that provides the expression for the modulation wave vector in the paramagnetic limit, $q^2 = -\beta/2\delta = 2K_3\left(T_{cP}\right)/K_5\left(T_{cP}\right)v_F^2 > 0$. This wave vector increases from 0 at the tricritical point to a very large value at $T \longrightarrow 0$.

So far, we have assumed that the absolute value of the FFLO wave vector might be determined in the paramagnetic limit only. However, the orbital contribution, in its turn, may influence the absolute value of the FFLO modulation vector. Making use of Equations (29) and (30), we may optimize its solution with respect to the vector **q**. The result of the calculations are depicted in Figure 11, which illustrates the modulus of the FFLO wave vector versus the reduced temperature. One can see that the orbital effect on the modulation vector is weak except for the region in the close proximity to the resonance, $q \approx Q$, where it may strongly influence the FFLO structure [88,89]. In the vicinity of the resonance, where the unperturbed $q$ and $Q$ curves intersect, an interesting lock-in effect appears: two wave vectors hybridize.

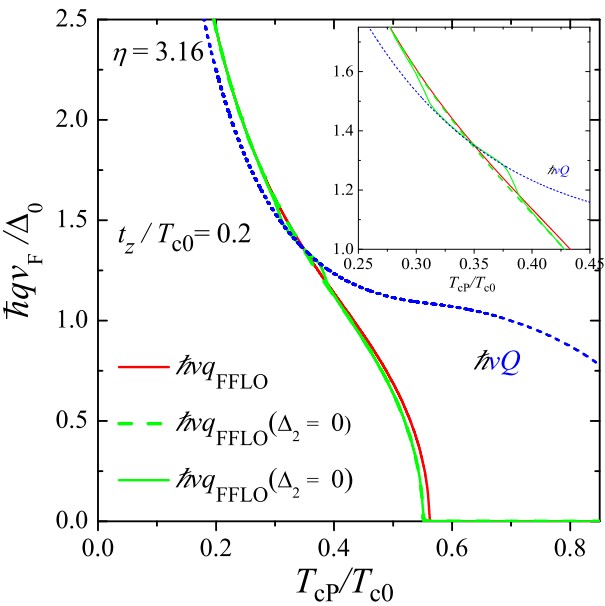

**Figure 11.** The absolute value of the FFLO modulation vector $q$ versus of $T_{cP}/T_{c0}$ for an s-wave quasi-1D superconductor when accounting for the orbital effects within the first iteration. Here, $t_c/T_c = 0.2$. The solid line takes into account the influence of resonance, and the short dashed line shows the field vector $Q$.

To investigate in this locking effect in detail, we apply the theory developed by Dzyaloshinskii [90–92] and look for the solution of Equation (35) in the following form $\Delta_0 \cos[Qx + \varphi(x)]$, where $\varphi(x)$ is a slowly varying function with $\varphi'(x) \ll Q$. This implies that the orbital effect weakly modifies the exact one-dimensional solution, which is justified for $(T_{c0} - T)/T_{c0} \gg (t/T_{c0})^2$. We insert this solution into the functional (36), perform the average over the period $Q^{-1}$ as $\langle ... \rangle = \frac{Q}{2\pi} \int_{-\pi/Q}^{\pi/Q} [...] \, dx$, and taking into account that $\varphi(x) \ll Qx$, obtain

$$
\begin{aligned}
\langle F_{sn} \rangle_Q = \frac{\Delta_0^2}{L_x} \int dx \frac{\alpha}{2} &+ \frac{1}{2}\beta \left[ Q + \varphi' \right]^2 \\
&+ \frac{1}{2}\delta \left[ Q + \varphi' \right]^4 + t^2\gamma \left[ 2 - \cos 2\varphi(x) \right] \\
&+ \frac{1}{8}t^2\mu Q^2 \left[ 2 - 7\cos 2\varphi(x) \right] \\
&+ \frac{6}{8}t^2\mu \left[ 2 - \cos 2\varphi(x) \right] \left[ Q + \varphi' \right]^2.
\end{aligned}
\tag{38}
$$

In the adopted approximation, the functional $F_{sn}$ depends only on the space varying phase, $\varphi(x)$. Since we are interested in the behavior of the system in the vicinity of the resonance $q \approx Q$, we can write the functional in a simpler form (normalized by $-\pi K_5(T_c)\Delta_0^2$, note that $K_5(T_c) < 0$)

$$
\begin{aligned}
\left\langle F_{sn}^\varphi \right\rangle_Q = \frac{\Delta_0^2}{L_x} \int dx \frac{-\alpha}{2} &+ \frac{1}{8}v_F^4 Q^2 \left( \varphi' - \delta q \right)^2 \\
&+ \frac{5}{8}t^2 v_F^2 Q^2 - \frac{11}{16}t^2 v_F^2 Q^2 \cos\left[ 2\varphi(x) \right],
\end{aligned}
\tag{39}
$$

where $\delta q \equiv Q - q$ is the detuning, or the relative misfit between two periodicities in the system. The phase $\varphi(x)$ is the shift of the FFLO modulation relative to the minima in the potential $\sim \cos[2\varphi(x)]$. The state $\varphi(x) = 0$ is the commensurate phase between $Q$ and $q$. Equation (39) is a continuum limit approximation of the Frenkel–Kontorova Hamiltonian introduced by Frank and Van der Merwe [93].

Taking into account the periodicity of the function $\varphi(x)$, we can obtain the ground state minimizing $\left\langle F_{sn}^{\varphi} \right\rangle_Q$, which is given by the solutions of the exactly integrable sine-Gordon equation for the phase $\varphi(x)$

$$2\frac{\partial^2 \varphi}{\partial x^2} + v \sin[2\varphi(x)] = 0, \tag{40}$$

where $v \equiv 11t^2/v_F^2$ is the effective interlayer coupling parameter. In the absence of this coupling, $v = 0$, Equation (40) has a solution $\varphi(x) = Cx$, which describes an unperturbed one-harmonic incommensurate phase, $\Delta_0 \cos(Q + C)x$. On the other hand, Equation (40) has a trivial solution, $\varphi = 0$, that corresponds to a commensurate structure, $\Delta_0 \cos Qx$. For a finite $v$, Equation (40) should describe some inhomogeneous distribution of the phase of the order parameter. The first integral of this equation is given by

$$\left(\frac{\partial \varphi}{\partial x}\right)^2 = \frac{v}{\kappa^2}\left\{1 - \kappa^2 \sin^2[\varphi(x)]\right\}, \tag{41}$$

where $\kappa^2 \equiv 2v/(\varepsilon + v)$ and $\varepsilon$ is the constant of integration. Then, the exact solution of Equation (41), expressed in terms of the Jacobi amplitude, is

$$\varphi(x) = \text{am}\left[\frac{\sqrt{v}}{\kappa}x, \kappa^2\right], \tag{42}$$

where $\kappa$ is the constant of integration and must be found from the energy minimum of the system. The solution is the regularly spaced solitons, a soliton lattice. The soliton lattice is a compromise between term $\cos[2\varphi(x)]$, which favors $\varphi(x) = \text{const}$ and the derivative part that favors $\varphi(x) = Cx$. For $\kappa \to 1$, it reduces to $\varphi(x) = 2\tan^{-1}\left[\tanh\frac{\sqrt{v}}{2\kappa}x\right]$ describing a domain wall, which separates two commensurate regions, one with phase $\varphi = -\pi/2$, and the other with $\varphi = \pi/2$. Within this wall (soliton), a $\pi$ change of the phase occurs. The length of the soliton can be a small meaning the fast change of the phase.

With the solution for $\varphi(x)$ given by Equation (42), the normalized energy of the state is expressed in terms of complete elliptic integrals $E$ an $K$ as

$$\left\langle F_{sn}^{\varphi} \right\rangle = \frac{\delta q}{4}\frac{\pi}{2}\frac{\sqrt{v}}{\kappa K(\kappa^2)} - \frac{v}{16}\left[1 - \frac{2}{\kappa^2} + 4\frac{E(\kappa^2)}{\kappa^2 K(\kappa^2)}\right]. \tag{43}$$

Minimization of the energy with respect to $\kappa$ leads to the following equation for $\kappa$ :

$$\frac{\kappa}{E(\kappa^2)} = \sqrt{\frac{v}{v_c}} \equiv \zeta, \tag{44}$$

where $v_c = \pi^2 \delta q^2/4$. Here, the parameter $\kappa$ varies in the range $0 \leqslant \kappa \leqslant 1$ as long as $v$ varies in the range $0 \leqslant v \leqslant v_c$. The variation of the parameter $\kappa$ results in a drastic changes in the behavior of the phase $\varphi(x)$. Figure 12 displays the phase for several values of parameter $\zeta \in \{0.2, 0.6, 0.8, 0.9, 0.99, 0.999999\}$. As $\zeta$ is increased a plateau section appears within the period of soliton lattice $L = 2K(\kappa^2)\kappa/\sqrt{v}$ or $\delta qL = (2/\pi)^2 E(\kappa^2)K(\kappa^2)$. With increasing $\zeta$, not only the period of the soliton lattice increases but also the relative fraction of the plateau width. On this plateau, the phase remains almost constant, but strongly changes beyond plateau at the end of the periods. Thus, the system can be represented as a periodic structure of domains of the locked phase separated by solitons. Equation (43) can be expressed differently via the distance $L$ between the domain walls, namely

$$\left\langle F_{sn}^{\varphi} \right\rangle = \frac{\pi}{4}\frac{\delta q}{L} - \frac{v}{16}\left[1 - \frac{2}{\kappa^2} + \frac{4}{\kappa^2}\frac{E(\kappa^2)}{K(\kappa^2)}\right]. \tag{45}$$

One can see that, as $\delta q \longrightarrow 0$, parameter $\zeta$ increases its value. In the limit $\zeta \longrightarrow 1$, the period of the soliton lattice diverges

$$\lim_{\zeta \to 1} L = \frac{2\kappa}{\sqrt{v}} \ln \frac{4}{\sqrt{1 - \kappa^2}}, \tag{46}$$

and the modulation wave vector is negligible. For $\zeta \geqslant 1$, there are no real valued solutions of Equation (44). Consequently, there exists a limiting value of $\delta q_c \equiv 2\sqrt{11}t/\pi v_F$, beyond which the phase is always locked, and the commensurate phase remains stable as $\delta q$ decreases further. This means that the modulation vector coincides with the modulation induced by the field. Therefore, at $\delta q_c$, we have a lock-in of the FFLO modulation to the field induced potential, i.e., incommensurate/commensurate type phase transition. This corresponds to $L \to \infty$ and the FFLO modulation is described by $\Delta_0 \cos Qx$. Figure 13 illustrates the behavior of the spatially varied order parameter in the vicinity of the incommensurate–commensurate (I–C) phase transition when $\zeta \to 1$. One can see the domain walls (phase solitons) in the spatial distribution of the order parameter.

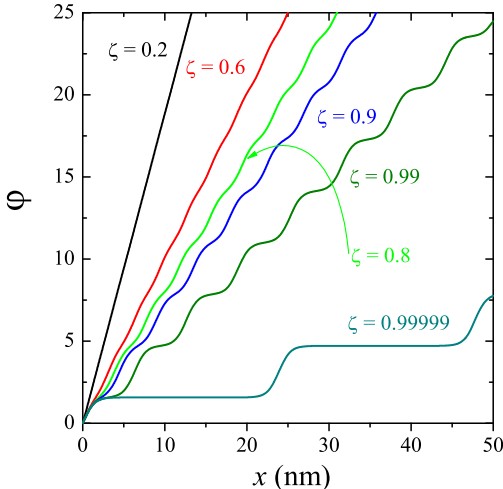

**Figure 12.** The phase shift of the FFLO modulation for several values of the anisotropy parameter $\zeta$. The straight line corresponds to an unperturbed incommensurate structure.

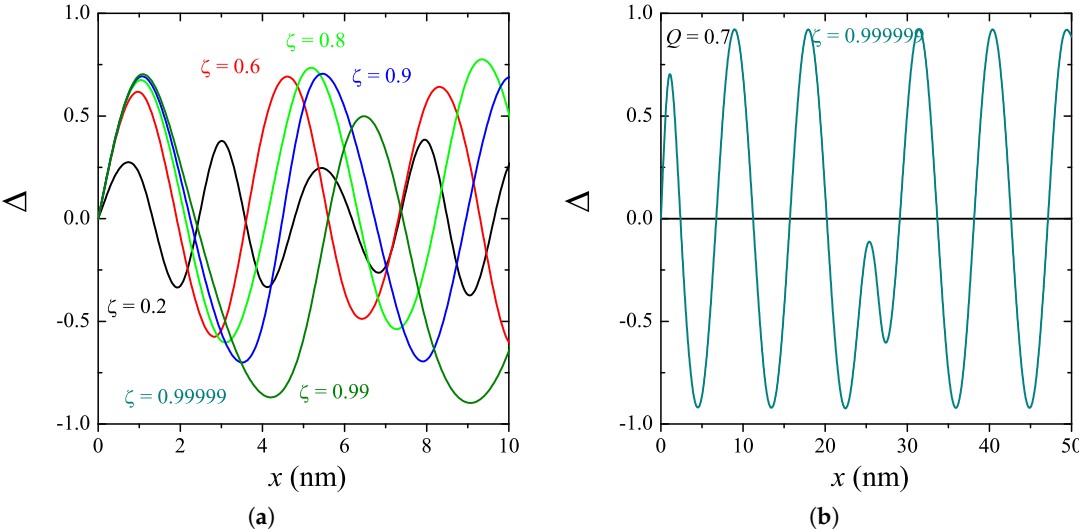

**Figure 13.** The spatial distribution of the order parameter in the vicinity of the I–C phase transition for $Q = 0.2$ and for several values of the anisotropy parameter $\zeta$ (**a**) and $Q = 0.7$ (**b**).

The theory developed is applicable not only to the quasi-1D systems but to the quasi-2D materials as well. Since the orientation of the FFLO modulation vector is determined by the crystal field effects [68,70] and/or the pairing symmetry [71], when the in-plane magnetic field is perpendicular to the FFLO wave vector **q** we reduce the quasi-2D case to the just described quasi-1D case, Label (36), but with the renormalized coefficients: $\beta \equiv \frac{\pi}{8} K_3(T) v_F^2$, $\delta \equiv -\frac{3\pi}{128} K_5(T) v_F^4$, $\mu \equiv -\frac{\pi}{4} K_5(T) v_F^2$, $\gamma \equiv \frac{\pi}{4} K_3(T)$. The coefficients $\gamma$, $\nu$ have the same values.

In this section, we have shown that, besides the anomalous cusps in the temperature and angular dependencies of the in-plane critical field, which can be considered as markers for the FFLO phase, a new marker is defined, which is based on the fact that the proximity to the resonance condition is characterized by the field-induced phase transition from the incommensurate to the commensurate structure (C–I) of the spatial modulation of the order parameter with respect to the magnetic wave vector structure. The periodic potential tends to "lock-in" the FFLO modulation into a commensurate configuration; in the incommensurate phase near the C–I transition, the periodicity of the phase solitons or phase-domain walls depends continuously on the temperature and external field strength.

## 6. Conclusions

In the past few decades, the quest to make evident the FFLO phase has stimulated a lot of activities in this branch of the field of superconductivity and considerably deepened our understanding of the physics behind. The FFLO concept is more than 50 years old and now we are probably at the beginning of its most interesting period, experimental evidence. The developed theoretical framework, both described or discussed/cited or even not mentioned in this review, are very timely to support the burst of the inhomogeneous phase experimental observation [39].

In this article, we have reviewed our contribution to the research field of inhomogeneous superconductivity and, in particular, we have shown that

- The FFLO modulation strongly interferes with the orbital effect and provides the main source of the in-plane critical field anisotropy. The change of the anisotropy of the critical field as well as of its fine structure may give important information about the FFLO state and unambiguously prove its existence.
- As soon as the vector potential of the applied magnetic field is commensurate with the wave vector of the FFLO phase, the resonance peaks appear in the field-direction dependence of the onset of superconductivity.
- At the resonance, the interplay between the orbital and paramagnetic effects may result in an FFLO lock-in effect.

These effects can open up new possibilities to an unambiguous evidence spatially modulated superconducting phase in quasi-low-dimensional conductors.

**Acknowledgments:** We acknowledge the support by the Agence Nationale de la Recherche (ANR) and the Deutsche Forschungsgemeinschaft (German Research Foundation, DFG) grant "Fermi-NESt".

**Author Contributions:** Both authors contributed equally to this work.

**Conflicts of Interest:** The authors declare no conflict of interest.

## Appendix A. Derivation of the Extended Nonlinear Lowerence–Doniach Equation

We start from the Eilenberger equations for a quasi-2D superconductor [1]

$$\left[ \Omega_n + \frac{1}{2} \widehat{\Pi}(\mathbf{Q}) \right] f_\omega(\mathbf{n_p}, \mathbf{r}, k_z, p_z) = \Delta(\mathbf{r}, k_z) g_\omega(\mathbf{n_p}, \mathbf{r}, k_z, p_z), \tag{A1}$$

$$\left[ \Omega_n - \frac{1}{2} \widehat{\Pi}(\mathbf{Q}) \right] f_\omega^\dagger(\mathbf{n_p}, \mathbf{r}, k_z, p_z) = \Delta^*(\mathbf{r}, k_z) g_\omega(\mathbf{n_p}, \mathbf{r}, k_z, p_z), \tag{A2}$$

$$\frac{1}{2}\widehat{\Pi}(0)\,g_\omega\left(\mathbf{n_p}, \mathbf{r}, k_z, p_z\right) = \Delta^*\left(\mathbf{r}, k_z\right) f_\omega\left(\mathbf{n_p}, \mathbf{r}, k_z, p_z\right) \tag{A3}$$
$$- f_\omega^\dagger\left(\mathbf{n_p}, \mathbf{r}, k_z, p_z\right)\Delta\left(\mathbf{r}, k_z\right),$$

where $\Omega_n \equiv \omega_n + 1/2\tau - ih\,\mathrm{sign}(\omega_n)$ and the definition of $\widehat{\Pi}$ is provided in Equation (3) or

$$\widehat{\Pi} = \mathbf{v}_F \nabla_{x,y} + i4t\sin\left(p_z d\right)\sin\left(\chi_A\right) \tag{A4}$$

with $\chi_A \equiv \mathbf{Qr} - \frac{k_z}{2}d$. We have assumed that the vector potential varies slowly at the inter-plane distances (this assumption means that we neglect the diamagnetic screening currents and take the magnetic field as uniform and given by the external field, *H*). These equations depend on $k_z$, which takes into account the dependence of the order parameter on the center-of-mass coordinate, $(z+z')/2$. The order parameter is defined self-consistently as

$$\frac{1}{\lambda}\Delta\left(\mathbf{r}, k_z\right) = 2\pi T\,\mathrm{Re}\sum_{\omega>0}\left\langle f_\omega\left(\mathbf{n_p}, \mathbf{r}, k_z, p_z\right)\right\rangle, \tag{A5}$$

where $\lambda$ is the pairing constant and the brackets definition is in Equation (7).

The derivation is performed iteratively. It is convenient to write the procedure as the following iterative scheme:

$$f_\omega^{(k+1)}\left(\mathbf{n_p}, \mathbf{r}, k_z, p_z\right) = \frac{\Delta}{\Omega_n}g_\omega^{(k)} - \frac{1}{2\Omega_n}\widehat{\Pi}f_\omega^{(k)}, \tag{A6}$$

$$f_\omega^{\dagger(k+1)}\left(\mathbf{n_p}, \mathbf{r}, k_z, p_z\right) = \frac{\Delta^*}{\Omega_n}g_\omega^{(k)} + \frac{1}{2\Omega_n}\widehat{\Pi}f_\omega^{\dagger(k)}. \tag{A7}$$

The zero order approximation (the absence of the superconducting state) results in $g_\omega^{(0)}\left(\mathbf{n_p}, \mathbf{r}, k_z, p_z\right) = 1$ and $f_\omega^{(0)}\left(\mathbf{n_p}, \mathbf{r}, k_z, p_z\right) = 0$. Then, for the first order approximation, the iterative scheme produces $f_\omega^{(1)}\left(\mathbf{n_p}, \mathbf{r}, k_z, p_z\right) = \Delta/\Omega_n$, $f_\omega^{\dagger(1)}\left(\mathbf{n_p}, \mathbf{r}, k_z, p_z\right) = \Delta^*/\Omega_n$. For the iterative scheme, we need the iterative expression for $g_\omega^{(k)}\left(\mathbf{n_p}, \mathbf{r}, k_z, p_z\right)$ function. If we write it, for example for $g_\omega^{(5)}$, as $g_\omega^{(5)} = \widetilde{g}_\omega^{(0)} + \lambda\widetilde{g}_\omega^{(1)} + \lambda_\omega^2\widetilde{g}_\omega^{(2)} + \lambda_\omega^3\widetilde{g}_\omega^{(2)} + \lambda^4\widetilde{g}_\omega^{(4)} + \lambda^5\widetilde{g}_\omega^{(5)}$ and $f_\omega^{(5)} = \widetilde{f}_\omega^{(0)} + \lambda\widetilde{f}_\omega^{(1)} + \lambda^2\widetilde{f}_\omega^{(2)} + \lambda^3\widetilde{f}_\omega^{(3)} + \lambda^4\widetilde{f}_\omega^{(4)} + \lambda^5\widetilde{f}_\omega^{(5)}$, then making use of the normalization condition

$$g_\omega^2 + f_\omega f_\omega^\dagger = 1, \tag{A8}$$

we obtain the following system of coupled expressions for the *k*-th iteration

$$\widetilde{g}_\omega^{(0)}\left(\mathbf{n_p}, \mathbf{r}, k_z, p_z\right) = 1, \tag{A9}$$

$$\sum_{i=0}^{k}\widetilde{g}_\omega^{(i)}\widetilde{g}_\omega^{(k-i)} = -\sum_{i=0}^{k}\widetilde{f}_\omega^{(k-i)}\widetilde{f}_\omega^{\dagger(i)}. \tag{A10}$$

The second iteration of the iteration scheme leads to

$$f_\omega^{(2)}\left(\mathbf{n_p}, \mathbf{r}, k_z, p_z\right) = \frac{\Delta\left(\mathbf{r}, k_z\right)}{\Omega_n} - \frac{1}{2\Omega_n^2}\widehat{\Pi}\Delta\left(\mathbf{r}, k_z\right), \tag{A11}$$

$$f_\omega^{\dagger(2)}\left(\mathbf{n_p}, \mathbf{r}, k_z, p_z\right) = \frac{\Delta^*\left(\mathbf{r}, k_z\right)}{\Omega_n} + \frac{1}{2\Omega_n^2}\widehat{\Pi}\Delta^*\left(\mathbf{r}, k_z\right). \tag{A12}$$

Therefore, here, $\widetilde{f}_\omega^{(2)} = -\widehat{\Pi}\Delta\left(\mathbf{r}, k_z\right)/2\Omega_n^2$, $\widetilde{f}_\omega^{\dagger(2)} = \widehat{\Pi}\Delta^*\left(\mathbf{r}, k_z\right)/2\Omega_n^2$. From the normalization condition, we can also obtain:

$$\widetilde{g}_\omega^{(1),2} + 2\widetilde{g}_\omega^{(0)}\widetilde{g}_\omega^{(2)} = -\widetilde{f}_\omega^{(2)}\widetilde{f}_\omega^{\dagger(0)} - \widetilde{f}_\omega^{(1)}\widetilde{f}_\omega^{\dagger(1)} - \widetilde{f}_\omega^{(0)}\widetilde{f}_\omega^{\dagger(2)}, \tag{A13}$$

$$2\widetilde{g}_\omega^{(0)} \widetilde{g}_\omega^{(2)} = -\widetilde{f}_\omega^{(1)} \widetilde{f}_\omega^{\dagger(1)}, \tag{A14}$$

which gives $\widetilde{g}_\omega^{(2)} = -\widetilde{f}_\omega^{(1)} \widetilde{f}_\omega^{\dagger(1)}/2 = -f_\omega^{(1)} f_\omega^{\dagger(1)}/2$ and $\widetilde{g}_\omega^{(1)} = 0$. Therefore, the normal Green function $g_\omega^{(2)}(\mathbf{n_p}, \mathbf{r}, k_z, p_z)$ acquires the following form

$$g_\omega^{(2)}(\mathbf{n_p}, \mathbf{r}, k_z, p_z) \quad = \quad \widetilde{g}_\omega^{(0)} + \widetilde{g}_\omega^{(1)} + \widetilde{g}_\omega^{(2)} \quad = \quad 1 - \frac{f_\omega^{(1)} f_\omega^{\dagger(1)}}{2} \quad = \quad 1 - \frac{|\Delta(\mathbf{r}, k_z)|^2}{2\Omega_n^2}. \tag{A15}$$

Following similar steps, then performing averaging over $p_y$ and $p_z$ and Fermi surface $v_{F_x}$, we obtain, on the 5th iteration for $f_\omega^{(5)} \equiv f_\omega^{(5)}((\mathbf{n_p}, \mathbf{r}, k_z, p_z))$,

$$
\begin{aligned}
f_\omega^{(5)} = & \frac{\Delta(\mathbf{r}, k_z)}{\Omega_n} - \frac{\Delta(x)|\Delta(\mathbf{r}, k_z)|^2}{2\Omega_n^3} + \frac{1}{4\Omega_n^3}\widehat{\Pi}^2\Delta(\mathbf{r}, k_z) + \frac{3\Delta(\mathbf{r}, k_z)|\Delta(\mathbf{r}, k_z)|^4}{8\Omega_n^5} \\
& + \frac{1}{16\Omega_n^5}\widehat{\Pi}^4\Delta(\mathbf{r}, k_z) - \frac{4}{8\Omega_n^5}|\Delta(\mathbf{r}, k_z)|^2\widehat{\Pi}^2\Delta(\mathbf{r}, k_z) - \frac{1}{8\Omega_n^5}\Delta^2(\mathbf{r}, k_z)\widehat{\Pi}^2\Delta^*(\mathbf{r}, k_z) \\
& - \frac{3}{8\Omega_n^5}\Delta^*(\mathbf{r}, k_z)\widehat{\Pi}\Delta(\mathbf{r}, k_z)\widehat{\Pi}\Delta(\mathbf{r}, k_z) - \frac{2}{8\Omega_n^5}\Delta(\mathbf{r}, k_z)\widehat{\Pi}\Delta(\mathbf{r}, k_z)\widehat{\Pi}\Delta^*(\mathbf{r}, k_z).
\end{aligned}
$$

Substituting the found averages into the self-consistency relation given by $\Delta^{(5)}(\mathbf{r}, k_z)/\lambda = \pi T \sum_n \left\langle f_\omega^{(5)}((\mathbf{n_p}, \mathbf{r}, k_z, p_z)) \right\rangle$ and using the standard regularization rule

$$\frac{1}{\lambda} = \ln\frac{T}{T_{c0}} + 2\pi T \sum_{\omega_n>0} \frac{1}{\omega_n}, \tag{A16}$$

we obtain the following extended version of the Ginzburg–Landau equation (for clarity, we dropped down $k_z$-dependence), if, by $\widehat{\Pi}$, one understands its 3D version,

$$
\begin{aligned}
0 = & \Delta(\mathbf{r})\left\{\pi\left[K_1(T) - K_1^0(T)\right] - \ln\frac{T}{T_{c0}}\right\} \\
& + \frac{\pi K_3(T)}{4}\widehat{\Pi}^2\Delta(\mathbf{r}) - \frac{\pi K_3(T)}{2}\Delta(\mathbf{r})|\Delta(\mathbf{r})|^2 \\
& + \frac{3\pi K_5(T)}{8}\Delta(\mathbf{r})|\Delta(\mathbf{r})|^4 + \frac{\pi K_5(T)}{16}\widehat{\Pi}^4\Delta(\mathbf{r}) \\
& - \frac{\pi K_5(T)}{8}\left\{4|\Delta(\mathbf{r})|^2\widehat{\Pi}^2\Delta(\mathbf{r}) + \Delta^2(\mathbf{r})\widehat{\Pi}^2\Delta^*(\mathbf{r})\right. \\
& \left. + 3\Delta^*(\mathbf{r})\widehat{\Pi}\Delta(\mathbf{r})\widehat{\Pi}\Delta(\mathbf{r}) + 2\Delta(\mathbf{r})\widehat{\Pi}\Delta(\mathbf{r})\widehat{\Pi}\Delta^*(\mathbf{r})\right\}, \tag{A17}
\end{aligned}
$$

where we have introduced the following notations [86] $K_m(T) = 2T \sum_{\omega>0} (\omega_n - i\mu_B H)^{-m}$ and $K_m^0(T) = 2T \sum_{\omega>0} \omega_n^{-m}$. This result is similar to the one obtained by Houzet and Mineev [13]. After the averaging procedure, we can write (neglecting the $k_z$ dependence)

$$
\begin{aligned}
\Delta(\mathbf{r})\ln\frac{T}{T_{c0}} = & \Delta(\mathbf{r})\pi\left[K_1(T) - K_1^0(T)\right] + \frac{\pi K_3(T)}{4}\partial^2\Delta(\mathbf{r}) - \frac{\pi K_3(T)}{2}\Delta(\mathbf{r})|\Delta(\mathbf{r})|^2 - 2t^2\pi K_3(T)\sin^2(\chi_A)\Delta(\mathbf{r}) \\
& + \frac{3\pi K_5(T)}{8}\Delta(\mathbf{r})|\Delta(\mathbf{r})|^4 + \frac{\pi K_5(T)}{16}\partial^4\Delta(\mathbf{r}) + \frac{t^2\pi K_5(T)}{4}\left[1 - 7\cos(2\chi_A)\right][\partial\chi_A]^2\Delta(\mathbf{r}) \\
& + 10t^2\pi K_5(T)\sin^2(\chi_A)\Delta(\mathbf{r})|\Delta(\mathbf{r})|^2 - t^2\pi K_5(T)\left[3\sin(2\chi_A)\partial\chi_A\partial\Delta(\mathbf{r})\right. \\
& \left. + \sin(2\chi_A)\partial^2\chi_A\Delta(\mathbf{r}) + 3\sin^2(\chi_A)\partial^2\Delta(\mathbf{r})\right] - \frac{\pi K_5(T)}{2}|\Delta(\mathbf{r})|^2\partial^2\Delta(\mathbf{r}) - \frac{\pi K_5(T)}{8}\Delta^2(\mathbf{r})\partial^2\Delta^*(\mathbf{r}) \\
& - \frac{3\pi K_5(T)}{8}\Delta^*(\mathbf{r})[\partial\Delta(\mathbf{r})]^2 - \frac{\pi K_5(T)}{4}\Delta(\mathbf{r})\partial\Delta(\mathbf{r})\partial\Delta^*(\mathbf{r}) + 6t^4\pi K_5(T)\sin^4(\chi_A)\Delta(\mathbf{r}). \tag{A18}
\end{aligned}
$$

Here, the derivative signs mean the following $\partial^2 \Delta(\mathbf{r}) = \left[ \langle v_{Fx}^2 \rangle \partial_x^2 + \langle v_{Fy}^2 \rangle \partial_y^2 \right] \Delta(\mathbf{r})$, $\partial^4 \Delta(\mathbf{r}) = \left[ \langle v_{Fx}^4 \rangle \partial_x^4 + \langle v_{Fx}^2 v_{Fy}^2 \rangle \partial_x^2 \partial_y^2 + \langle v_{Fy}^4 \rangle \partial_y^4 \right] \Delta(\mathbf{r})$, $\partial \chi_A \partial \Delta(\mathbf{r}) = \langle v_{Fx}^2 \rangle \partial_x \chi_A \partial_x \Delta(\mathbf{r}) + \langle v_{Fy}^2 \rangle \partial_y \chi_A \partial_y \Delta(\mathbf{r})$ and $[\partial \chi_A]^2 = \langle v_{Fx}^2 \rangle [\partial_x \chi_A]^2 + \langle v_{Fy}^2 \rangle [\partial_y \chi_A]^2$. In a quasi-1D case, this equation reads as

$$
\begin{aligned}
\alpha \Delta(x) = {} & \beta \partial_x^2 \Delta(x) - \delta \partial_x^4 \Delta(x) - 2\gamma \Delta(x) |\Delta(x)|^2 - 3\nu \Delta(x) |\Delta(x)|^4 + \mu |\Delta(x)|^2 \partial_x^2 \Delta(x) \\
& + \tfrac{\mu}{4} \Delta^2(x) \partial_x^2 \Delta^*(x) + \tfrac{3\mu}{4} \Delta^*(x) [\partial_x \Delta(x)]^2 + \tfrac{\mu}{2} \Delta(x) \partial_x \Delta(x) \partial_x \Delta^*(x) - \tfrac{t^2 \mu}{2} [1 - 7 \\
& \times \cos(2\chi_A)] [\partial_x \chi_A]^2 \Delta(x) - 80 t^2 \nu \sin^2(\chi_A) \times \Delta(x) |\Delta(x)|^2 + 2 t^2 \mu [3 \sin(2\chi_A) \partial_x \chi_A \partial_x \Delta(x) \\
& + \sin(2\chi_A) \partial_x^2 \chi_A \Delta(x) + 3 \sin^2(\chi_A) \partial_x^2 \Delta(x)] - 8 t^2 \gamma \sin^2(\chi_A) \Delta(x) - 48 t^4 \nu \sin^4(\chi_A) \Delta(x),
\end{aligned} \tag{A19}
$$

where $\alpha \equiv \ln \frac{T}{T_{c0}} - \pi \left[ K_1(T) - K_1^0(T) \right]$, $\beta \equiv \frac{\pi}{4} K_3(T) \langle v_{Fx}^2 \rangle$, $\delta \equiv -\frac{\pi}{16} K_5(T) \langle v_{Fx}^4 \rangle$, $\gamma \equiv \frac{\pi}{4} K_3(T)$, $\mu \equiv -\frac{\pi}{2} K_5(T) \langle v_{Fx}^2 \rangle$, $\nu \equiv -\frac{\pi}{8} K_5(T)$. Here, the derivative signs mean the following $\partial^2 \Delta(x) = \langle v_{Fx}^2 \rangle \partial_x^2 \Delta(x)$, $\partial^4 \Delta(x) = \langle v_{Fx}^4 \rangle \partial_x^4 \Delta(x)$, $\partial \chi_A \partial \Delta(x) = \langle v_{Fx}^2 \rangle \partial_x \chi_A \partial_x \Delta(x)$ and $[\partial \chi_A]^2 = \langle v_{Fx}^2 \rangle [\partial_x \chi_A]^2$. The linearized version of this equation,

$$
\begin{aligned}
\alpha \Delta(x) = {} & \beta \partial_x^2 \Delta(x) - \delta \partial_x^4 \Delta(x) - 48 t^4 \nu \sin^4(\chi_A) \Delta(x) - \tfrac{t^2}{2} \mu [1 - 7 \cos(2\chi_A)] [\partial_x \chi_A]^2 \Delta(x) \\
& + + 2 t^2 \mu [3 \sin(2\chi_A) \partial_x \chi_A \partial_x \Delta(x) + 3 \sin^2(\chi_A) \partial_x^2 \Delta(x)] - 8 t^2 \gamma \sin^2(\chi_A) \Delta(x)
\end{aligned} \tag{A20}
$$

is used in Section 5 of the review.

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
