# Peer review of "In Search of Unambiguous Evidence of the Fulde–Ferrell–Larkin–Ovchinnikov State in Quasi-Low Dimensional Superconductors"

_condensedmatter, doi:10.3390/condmat2030030_

Round 1

Reviewer 1 Report

In this article, the authors present a comprehensive review of their predictions for the dependence of the spatially modulated  "FFLO" spatially modulated superconducting state on the orientation of an applied magnetic field within the superconducting plane for layered superconductors (rather than as a function of angle rotating out of the superconducting plane), along with an  introductory review of the growing macroscopic and microscopic experimental evidence  for this FFLO phase.   As such,  the analysis presented in this article makes a unique and valuable contribution towards our understanding of  the properties of the FFLO phase (even as a review article) as much of other theoretical  work on this unusual superconducting phase focuses on the effects of rotation of the magnetic field out of the superconducting plane, the introduction of magnetic vortices, and the destruction of the FFLO phase. 

The authors assert that their predicted dependence of the superconducting critical temperature and upper critical field on field angle for magnetic fields within the plane can provide unambiguous evidence of the spatially modulated superconducting phase in quasi-low dimensional layered conductors.  Experimentally, it is now established that the FFLO phase is only seen for fields parallel or nearly so with respect to the superconducting layers, but little experimental work has been done for rotations within the plane while in the FFLO phase, making the predictions of renewed experimental as well as theoretic interest. Nevertheless, these would be difficult experiments to carry out. The unforgivingly strong angle dependence of the FFLO phase for rotations out of the plane means that caution would be needed not to confuse a small wobbling in and out of the plane as the crystal or field is rotated with an angular dependent effect within the plane itself, and the authors may want to provide guidance in a follow up paper as to how to distinguish between the two effects. 

I am not in a position to evaluate the theoretical validity of the calculations but do believe that the theoretical work presented here will provide a useful test for experiment and an incentive to carry out more experiments of the in-plane field dependence.  With regards to the title, a quibble might be made as to whether this provides "direct evidence" of the FFLO state (for which microscopic NMR and macroscopic specific heat measurements now exist,  along with other supporting measurements such as magnetic torque, thermal conductivity, and rf-penetration depth), as noted by the authors. For me, direct evidence would be the imaging of the spin and charge distribution of the electrons revealing the formation of  paramagnetic spin domains at the nodes of the FFLO superconducting order parameter. As "a search for direct evidence," however, the title does appear to be technically valid, and the authors' main point is that evidence other than direct imaging might also reveal the modulation of the superconducting order parameter. 

In the draft version provided for review, I find a couple of errors and instances of confusing labeling in the figures and citations of experimental work that should be cleaned up before publication. I list these below. I recommend acceptance after minor revision to address these issues. 

________

In figure 1, the figure does not make clear to which Fermi surface --- the red spin-up surface or the blue spin down surface --- the labels kF-Δk and kF+Δk apply.  Also, the presumption is that the magnetic field is directed upwards, but there is no indication to that effect. Perhaps the labels could have arrows pointing to the respective surface, or be color coded in the same way as the corresponding Fermi surfaces. It may be that the authors consider this obvious, but then why the labels?  An arrow for the direction of the magnetic field could similarly clarify. 

In figure 2, the x-y-z axes represent a left-handed coordinate system, but the mathematics provided for the vector potential appears to imply a traditional right handed coordinate system. This could be solved most simply by swapping the x and y labels on the coordinate axes. Also, since the key point of the article is the dependence of physical quantities on the orientation of the magnetic field within the superconducting layers, it would be more instructive to show a magnetic field angle that wasn't directly along y (or - y) but instead at an angle other than 90 degrees to the x axis. This would also allow for inclusion of a symbol for the angle α  (alpha) illustrating  its meaning in the accompanying text. 

Finally, the intriguing first order phase line reported in reference 22 based on specific heat measurements is no longer considered to correspond to the FFLO phase boundary but instead to an interesting vortex effect for small rotations of the magnetic field out of the superconducting planes, as noted in references 23 and 31 by the same research group. This affects lines 65 - 67 of the draft provided for review (which would need to be reworded in some way). 

Author Response

We are grateful to the referee for the evaluation of our manuscript and thoughtful suggestions helping us to improve the presentation of results. Below is our response to the specific comments.

Q-1)

In figure 1, the figure does not make clear to which Fermi surface --- the red spin-up surface or the blue spin down surface --- the labels kF-Δk and kF+Δk apply.  Also, the presumption is that the magnetic field is directed upwards, but there is no indication to that effect. Perhaps the labels could have arrows pointing to the respective surface, or be color coded in the same way as the corresponding Fermi surfaces. It may be that the authors consider this obvious, but then why the labels?  An arrow for the direction of the magnetic field could similarly clarify. 

Reply: Figure 1 has been modified accordingly.

Q-2)

In figure 2, the x-y-z axes represent a left-handed coordinate system, but the mathematics provided for the vector potential appears to imply a traditional right handed coordinate system. This could be solved most simply by swapping the x and y labels on the coordinate axes. Also, since the key point of the article is the dependence of physical quantities on the orientation of the magnetic field within the superconducting layers, it would be more instructive to show a magnetic field angle that wasn't directly along y (or - y) but instead at an angle other than 90 degrees to the x axis. This would also allow for inclusion of a symbol for the angle α  (alpha) illustrating  its meaning in the accompanying text. 

Reply: Figure 2 has been modified accordingly

Q-3:

Finally, the intriguing first order phase line reported in reference 22 based on specific heat measurements is no longer considered to correspond to the FFLO phase boundary but instead to an interesting vortex effect for small rotations of the magnetic field out of the superconducting planes, as noted in references 23 and 31 by the same research group. This affects lines 65 - 67 of the draft provided for review (which would need to be reworded in some way). 

Reply: Lines 65-67 are modified: "the calorimetric and" is removed, Line 75 is modified.

Reviewer 2 Report

The authors present a thorough, comprehensive review of their previously published theoretical work on the emergence of spatially inhomogeneous superconducting (SC) phases, the "FFLO state", in layered organic SCs. In particular, they delineate their proposal of a set of observations that could serve as clear-cut evidence for the experimental realization of FFLO-like phases.

The introductory section sets the stage perfectly, providing both a short but concise overview of the physics behind the FFLO phase and a review of relevant recent advances, theoretical and experimental. The subsequent sections give an extensive, well-organized presentation of the theoretical framework devised, and previously published in several papers, by the authors.

Overall, this is an excellent review article, which can be useful and relevant to both experts and novices in the field of unconventional superconductivity.

Some minor points to be considered:

-- The abstract should be slightly modified/rephrased: Lines 1-5 overlap, verbatim, with part of the introduction in Phys. Rev. Lett.108, 207005 (2012); Moreover, lines 7-11 are identical to fraction of the abstract in Phys. Rev. B 94, 214512 (2016).

-- The language could be polished a bit throughout. Punctuation is at times misleading, articles are missing, etc.

Author Response

We thank the Referee  for her/his careful reading of our manuscript, and her/his recommendation for publication of our paper in MDPI Condensed Matter  provided some issues are addressed

Q-1:

The abstract should be slightly modified/rephrased: Lines 1-5 overlap, verbatim, with part of the introduction in Phys. Rev. Lett.108, 207005 (2012); Moreover, lines 7-11 are identical to fraction of the abstract in Phys. Rev. B 94, 214512 (2016).-- The language could be polished a bit throughout. Punctuation is at times misleading, articles are missing, etc.

The abstract is refined and the text is polished.
